# Exploring team dynamics and performance in extended space missions using agent-based modeling

Iser Pena, Hao Chen*

Department of Systems and Enterprises, Stevens Institute of Technology, Hoboken, New Jersey, United States of America

* hao.chen@stevens.edu

## Abstract

As humanity prepares for crewed missions to Mars, these will be among the most extended and isolated journeys, presenting challenges in team performance and dynamics. The extreme isolation, prolonged confinement, and necessity for autonomous decision-making without Earth support demand a comprehensive understanding of crew interactions and psychological resilience. This study addresses this need by integrating psychological theories with agent-based modeling (ABM) to simulate the impact of team composition over a 500-day Mars mission. Utilizing the Five-Factor Model (FFM) of personality traits, agents varying in openness, conscientiousness, neuroticism, extraversion, and agreeableness were modeled to form both heterogeneous and homogeneous teams. To capture functional as well as psychological diversity, the framework also incorporates variation in skills and roles, enabling a 2×2 factorial design that disentangles the effects of personality heterogeneity, role heterogeneity, and their interaction. Our results suggest that team composition influences stress, health, performance, and cohesion, with personality variation and role specialization contributing distinctly. When combined, these factors appear to generate interaction effects relevant for crew selection, training, and preparation. These findings underscore the importance of jointly considering personality and functional diversity when developing strategies to support resilience and effectiveness in long-duration spaceflight.

## 1 Introduction

The ambition to explore Mars represents a pivotal moment in space exploration, pushing the boundaries of human endurance and cooperation. These missions are expected to involve unprecedented durations, potentially spanning two to three years [1], during which astronauts will confront extreme isolation and confinement while navigating the vast distance from Earth. One of the foremost challenges is the significant communication delay with signals taking up to 22 minutes one-way between Earth and Mars [2] which precludes real-time communication and compels crews to operate with a high degree of autonomy, making critical decisions independently.

**Data availability statement:** All relevant data are within the manuscript.

**Funding:** The author(s) received no specific funding for this work.

**Competing interests:** No authors have competing interests

The cumulative stress resulting from prolonged confinement in limited living spaces and heightened responsibility can lead to psychological effects such as mood disturbances, decreased motivation, and interpersonal conflicts [3,4]. The absence of personal space and privacy exacerbates stress levels, potentially affecting team cohesion and performance [5]. Additionally, the inability to receive prompt rescue or resupply intensifies perceived risks, impacting psychological resilience and necessitating robust coping mechanisms [6]. Understanding and optimizing team dynamics under these extreme conditions is crucial, as effective collaboration, stress management, and psychological support systems are essential for the success of long-duration space missions [7].

As mission durations extend into deep space, sustaining crew performance requires more than technical skill, it hinges on psychological adaptation, the dynamic process through which individuals adjust their thoughts, behaviors, and emotions in response to prolonged isolation, operational stress, and limited social interaction. This adaptive capacity is not random, it is strongly shaped by stable personality traits, particularly those outlined in the FFM [8]. These traits influence how individuals respond to stress, maintain motivation, and interact within a small, confined team over time. Framing psychological adaptation as an expression of personality under chronic stress provides a more nuanced understanding of astronaut behavior, enabling us to anticipate how different personality compositions may influence team functioning. Integrating this perspective into selection, training, and mission planning, space agencies can proactively support resilience, cohesion, and long-term performance in the extreme conditions of Mars-class missions.

With NASA's Artemis missions now underway and crewed Mars missions becoming increasingly concrete, there is a timely and critical need to develop predictive tools capable of assessing and optimizing team composition, psychological resilience, and operational effectiveness under realistic, Mars-like conditions. Existing studies have primarily relied on shorter-duration missions or terrestrial analog environments, such as the ISS (International Space Station), Mir, Mars-500, and HI-SEAS (Hawai'i Space Exploration Analog and Simulation), which do not fully capture the cumulative psychological and operational complexities of multi-year Mars expeditions [9,10]. Our study responds to this need by modeling how personality traits influence stress responses and resilience across long-duration missions. We interpret these dynamics as psychological adaptation, not as an isolated construct, but as the patterned ways in which traits shape coping and performance under chronic stress. Rather than focusing solely on trait-based outcomes or environmental triggers, we adopt a systems perspective that captures their dynamic interplay across time. This contribution supports the development of targeted interventions for crew selection, cohesion training, and resilience building protocols essential to the success of future deep-space exploration.

While ABM offers potential for simulating complex team interactions, existing frameworks remain insufficiently developed for space exploration contexts. For example, Devia et al. introduced graphical techniques to visualize how psychological traits influence social interactions in ABMs [11], yet their application to long-duration missions under compounded stressors is largely unexplored. Similarly, Sznajd-Weron et al. demonstrated how assumptions about individual traits versus situational factors dramatically impact model outcomes, emphasizing the importance of integrating both dimensions [12]. Despite these advances, there remains a lack of comprehensive models that address the intricate interplay between individual characteristics, team dynamics, and extreme environmental factors over prolonged durations.

Beyond individual resilience, effective mission functioning also depends on how skills and responsibilities are distributed across the crew. Evidence from spaceflight and analog environments suggests that redundant or overlapping roles can create inefficiencies, whereas

complementary expertise helps sustain operations under isolation and delayed communication [6,13]. Analyses of multiteam systems further point to role differentiation and task management as essential coordination mechanisms when crews must operate with high autonomy [14].

Despite this recognition, computational studies have rarely examined personality heterogeneity and functional specialization together. Most agent-based work isolates either personality variation or task-based diversity, leaving their combined influence underexplored [15].

This study addresses that need by formalizing both dimensions within a unified $2 \times 2$ factorial design, enabling direct comparison of their separate and interactive contributions during simulated Mars-class missions. Our research introduces a novel ABM framework that incorporates the Five-Factor Model (FFM) of personality traits to simulate team dynamics during extended space missions. By capturing the effects of personality diversity on stress, health, performance, and team cohesion, this framework provides a detailed understanding of how heterogeneous and homogeneous teams function under Mars-like conditions. The insights gained from this research support the development of strategies to enhance resilience, collaboration, and mission success in the demanding context of long-duration space exploration.

## 1.1 Literature review

**Team dynamics in space missions.** There has been consistent recognition that team dynamics play a critical role in determining the success of space missions, particularly given the extreme psychological and operational demands of isolated, confined, and autonomous environments. Bell et al. [16] provide one of the most comprehensive assessments of this topic, emphasizing that team composition is not only about individual competence but also about the configuration of interpersonal compatibility that influences both social integration and team performance. Their review identifies two major pathways through which composition affects mission outcomes: (1) by shaping the quality of interpersonal relations such as cohesion, conflict, and subgroup formation and (2) by affecting critical team processes such as coordination, shared cognition, and adaptability. The report also highlights that both surface-level (e.g., age, sex, culture) and deep-level attributes (e.g., personality traits, values, dominance) can lead to dysfunction when not aligned or managed, particularly in environments requiring high interdependence and limited external support. These insights underscore the importance of proactive team design and compatibility assessments in spaceflight planning.

Cohesion defined as the strength of social bonds and shared commitment among crew members has emerged as a core factor in mission stability. Rather than a static trait, it evolves over time and is shaped by interpersonal compatibility, communication style, and group adaptation. Analog missions such as Mars-500 and HI-SEAS have shown that cohesion tends to decline or fluctuate during periods of stress, especially mid-mission [17,18]. These cohesion patterns reflect how interpersonal dynamics adapt under isolation and stress, emphasizing that team composition directly influences the crew's psychological resilience [16].

Research on the ISS missions has also revealed the psychological challenges faced by astronauts in prolonged spaceflight. Kanas et al. [7] conducted studies involving surveys of astronauts and cosmonauts during and after missions to assess interpersonal relations and psychological well-being. Results indicated that interpersonal tensions, such as misunderstandings and conflicts, were common and could negatively impact crew morale and performance. The importance of effective communication, cultural understanding, and conflict resolution skills was emphasized as crucial in mitigating these issues.

Similarly, during the Mir space station missions, crew members experienced significant interpersonal stress. A notable incident involved a serious conflict between Russian cosmonauts and a visiting astronaut, leading to communication breakdowns and operational inefficiencies [19]. These events underscored the challenges of multicultural crews and the potential for conflicts arising from cultural differences.

Mars analog missions, such as the Mars-500 project and the HI-SEAS, have provided valuable insights into team dynamics during long-duration isolation [17,18]. In the Mars-500 project, six crew members were confined for 520 days to simulate a mission to Mars. Researchers observed that over time, the crew experienced declines in mood, increased interpersonal tension, and reduced motivation. Notably, cohesion fluctuated, with periods of increased conflict, especially during the mid mission phase [17]. In the HI-SEAS missions, crews lived in a dome on a remote volcanic site in Hawai'i for periods ranging from four to twelve months. Analyses revealed that crew cohesion and individual well-being were significantly influenced by personality compatibility and communication styles [18]. Effective leadership and clear communication were identified as critical factors in maintaining team harmony.

Antarctic expeditions serve as terrestrial analogs for space missions due to their extreme environments and isolation. Studies have documented phenomena such as the "third-quarter phenomenon," where psychological distress peaks during the middle of the mission [20]. Interpersonal conflicts, mood disturbances, and declines in cognitive function have been reported, emphasizing the need for psychological support and team-building interventions [21].

Recent contributions from organizational and communication sciences have emphasized the importance of conceptualizing spaceflight teams not merely as cohesive units, but as adaptive systems embedded within evolving task environments. Lungeanu et al. [14] adopt a task management perspective to analyze multiteam systems in spaceflight, revealing how coordination patterns shift under conditions of increasing crew autonomy, temporal pressure, and dynamic operational priorities. Their findings underscore the need to understand how team structures evolve as astronauts assume responsibilities traditionally managed by Earth-based support. Building on this, Antone et al. [22] highlight the value of computational approaches to examine how individual differences, such as personality traits, interact with team composition and influence performance over time. They call for integrative frameworks capable of capturing nonlinear, time-sensitive team processes. Similarly, Mesmer-Magnus et al. [15] identify key complexities unique to spaceflight teams including delayed communication, adaptive role shifts, and cumulative psychological stressors and advocate for a new generation of theoretical and modeling tools to guide crew design and intervention strategies. Together, these perspectives provide a conceptual foundation for exploring more dynamic and empirically grounded representations of team functioning in spaceflight.

**1.1.1 Psychological adaptation in extreme environments.** Understanding psychological adaptation and team dynamics in extreme environments is essential for improving crew performance and mission success. The confined and isolated settings of space missions pose unique psychological challenges that require comprehensive strategies to address. Researchers have explored various theoretical frameworks to explain how individuals and teams cope with the stressors inherent in such settings, integrating these perspectives with empirical findings from space missions and analog environments.

Environmental psychology examines the interplay between individuals and their physical surroundings, focusing on how confinement, isolation, and sensory deprivation impact psychological well-being and interpersonal relations. Environmental and situational factors, such

as limited space, lack of privacy, and monotonous surroundings, can affect mood and behavior [5]. Designing habitats that consider human factors and provide opportunities for recreation and personal space can improve psychological well-being. Incorporating elements that simulate natural environments, providing varied lighting conditions, and ensuring ergonomic design can alleviate some of the stress associated with long-term confinement [23].

Cognitive strategies and coping mechanisms play a crucial role in managing stress in these environments. Individuals employ various cognitive strategies to handle stress, such as positive reinterpretation, problem-solving, and acceptance [24]. Effective coping mechanisms enhance psychological resilience, allowing crew members to maintain performance under stress. Teams that foster adaptive coping can mitigate the adverse effects of isolation and confinement. Access to psychological support, both through pre-mission training and during the mission, helps in managing stress and maintaining mental health [7]. Interventions may include stress management techniques, counseling resources, and promoting social support within the team. Regular psychological assessments can identify issues early, allowing for timely interventions [21].

Cultural and language differences may pose additional challenges in multicultural crews. Language barriers and differing cultural norms can lead to misunderstandings and conflicts [25,26]. Emphasizing cultural competence, inclusive practices, and mutual respect can mitigate these issues and enhance team cohesion. Crew selection processes can focus on identifying individuals with compatible personality traits and strong coping abilities to reduce the likelihood of conflicts and improve overall team functioning [27]. Pre-mission training that includes communication skills, cultural awareness, and language skills can prepare crew members for the diversity they will encounter [28].

Communication patterns are closely linked to team cohesion and performance. Effective communication, characterized by regular and open exchanges, can prevent misunderstandings and build trust among crew members [29,30]. Establishing clear communication protocols, utilizing reliable communication technologies, and promoting active listening are essential for mission success.

Leadership style and clarity of roles within the team significantly impact team dynamics. Participative leadership and well-defined roles have been associated with higher team effectiveness [18]. Leaders who are adaptable, culturally sensitive, and supportive can positively influence team morale and performance. Clear role definitions ensure that each team member understands their responsibilities, reducing ambiguity and potential conflicts.

The integration of theoretical perspectives with practical applications is essential for the development of strategies aimed at enhancing team dynamics. Considerations related to environmental design such as the inclusion of private spaces, recreational facilities, and aesthetically pleasing elements can significantly reduce the adverse effects of confinement and monotony [23]. Furthermore, the provision of continuous psychological support throughout the duration of the mission is critical for sustaining mental health and promoting team cohesion [21].

**1.1.2 Personality traits.** A key factor in team dynamics and performance are personality traits, especially in high-stress and isolated environments like space missions. These traits fundamentally shape how individuals perceive and respond to prolonged stress, confinement, and interpersonal friction conditions that cannot be fully mitigated through training alone. In such extreme environments, where support from Earth is delayed and crews must operate autonomously, personality driven behaviors like emotional regulation, adaptability, and social sensitivity become critical for maintaining cohesion and operational stability [7,31].

The FFM, also known as the Big Five personality traits Openness to Experience, Conscientiousness, Extraversion, Agreeableness, and Neuroticism provides a comprehensive framework for understanding individual differences [8]. It enables mission planners to anticipate stress responses, communication dynamics, and interpersonal compatibility, offering predictive value for selecting and supporting crews capable of sustaining long-duration missions [33].

1. **Conscientiousness**: Associated with reliability, responsibility, and a strong work ethic. Barrick and Mount [34] found that conscientiousness positively correlates with job performance across various occupations, including teamwork settings.
2. **Agreeableness**: Linked to cooperation, compassion, and good naturedness. High agreeableness in team members correlates with better interpersonal relations and reduced conflict [35].
3. **Neuroticism**: Indicates emotional instability and anxiety. Individuals high in neuroticism may be more susceptible to stress and negatively impact team morale [36].
4. **Extraversion**: Characterized by sociability and assertiveness. Extraverts can contribute to positive team interactions but may dominate discussions if not balanced [37,38].
5. **Openness to Experience**: Involves creativity and adaptability. Teams with members high in openness may excel in problem-solving and adapting to new situations [39].

Musson et al. [31] examined the personality profiles of astronauts and found that successful crew members often scored high in conscientiousness and agreeableness, and low in neuroticism. These traits contributed to better stress management and interpersonal relations during missions. Kanas [19] emphasized the importance of selecting crew members with compatible personalities to minimize conflicts and enhance team cohesion. They noted that individual differences in coping styles and stress responses can significantly affect team dynamics.

Research has investigated how the composition of personality traits within a team affects performance. Peeters et al. [38] found that teams with lower variability in conscientiousness and agreeableness performed better due to shared work values and cooperative behaviors. Conversely, diversity in certain traits like openness can enhance creativity and problem-solving. A balance between homogeneity in traits that promote cohesion and diversity in traits that foster innovation is suggested to optimize team performance [40]. In addition to psychological factors, several studies point to the operational importance of skill diversity in spaceflight teams. For example, Sandal et al. [17] found in a 105-day space simulation that mismatches in functional responsibilities could create stress and inefficiencies, while balanced specialization improved perceived compatibility. Bishop et al. [18] likewise emphasized that clear differentiation of tasks and expertise supported stable performance in simulated long-duration crews. More broadly, reviews of team research in space and analog environments argue that composition must be understood across multiple dimensions, including both psychological compatibility and functional specialization [15]. However, most computational studies of spaceflight teams continue to model only one of these forms of heterogeneity at a time, leaving open questions about how personality diversity and role allocation interact in shaping stress, cohesion, and performance.

Traditional empirical and analog studies have provided valuable insights into stress, adaptation, and personality in spaceflight, but they remain limited in scope. Most capture only short durations or isolated factors, making it difficult to study the nonlinear and cumulative dynamics of multi-year missions [7,9]. Moreover, they cannot easily integrate individual heterogeneity (e.g., stable personality traits) and system-level processes (e.g., feedback loops

between stress, cohesion, and performance). Computational modeling approaches provide a way to address these challenges by enabling controlled, repeatable experiments on complex social systems [41]. In particular, Agent-Based Modeling (ABM) offers a framework for linking micro-level variation among individuals to emergent team-level outcomes across time. Building on these limitations, the next section considers how ABM has been applied to study team dynamics and performance in spaceflight and other extreme environments.

### 1.1.3 ABM for investigating team dynamics and performance.

ABM is a computational methodology that simulates the actions, interactions, and decision-making processes of autonomous agents within a defined environment. These agents representing individuals, groups, or organizations are endowed with distinct attributes and behavioral rules [41,42]. By allowing agents to interact according to relatively simple local rules, ABM can yield complex, emergent patterns at the system level that may not be deducible from the characteristics of any single agent [43]. ABM provides a bottom-up approach to understanding how collective phenomena arise from individual actions. A major strength of ABM lies in its capacity to incorporate heterogeneity at the individual level. Since each agent can be assigned unique traits, preferences, and rules of interaction, it is possible to capture nuanced variations in behaviors and responses across a population. Through iterative simulations, ABM enables researchers to examine how differences in agent attributes influence emergent dynamics, revealing patterns not readily discerned using more traditional analytical or equation based modeling techniques. Applications have spanned a variety of domains ranging from crowd behavior and social network analysis to market trends and information diffusion demonstrating ABM's flexibility and utility for understanding complex, adaptive systems [44].

Beyond broad social or economic systems, ABM has proven valuable in examining the internal dynamics of teams and workgroups, particularly when individual differences and personality traits are significant factors. By modeling agents as team members with distinct cognitive capabilities, communication patterns, or motivational profiles, ABM allows researchers to explore how these differences shape overall group outcomes.

Van Veen [45] applied ABM to study how various information aggregation mechanisms influence group decision-making. Agents were equipped with human-like reasoning abilities and exposed to different communication strategies, cognitive constraints, and levels of informational diversity. The findings revealed that approaches encouraging comprehensive information sharing such as random information dissemination and instructed dissent substantially improved decision quality. This underscores the importance of communication protocols and information flow in steering teams toward more effective outcomes.

In another study, Bergner et al. [46] used ABM to investigate how individual cognitive and noncognitive traits such as talkativeness, agreeableness, and critical thinking affected team performance. Their simulations showed that teams with higher average agreeableness and talkativeness achieved greater consensus and demonstrated improved collective performance. Although an increase in critical thinking initially reduced consensus, it still contributed modestly to performance improvements. These results highlight the delicate balance between fostering agreeable, communicative team environments and encouraging critical, rigorous thinking.

Perišić [47] further illustrated ABM's capacity to evaluate how diversity in cognitive processes and interaction styles supports team adaptation. In simulating product development teams facing evolving challenges, the study found that more cognitively diverse teams were better at learning, adjusting strategies, and sustaining performance under changing conditions. Such insights emphasize the inherent value of cultivating a variety of cognitive profiles and communication behaviors within teams to enhance resilience and flexibility.

**1.1.4 Applying ABM to long-duration space missions.** The complexity of long-duration space missions poses unique challenges for both individual crew members and the team as a whole. Factors such as prolonged isolation, confined habitats, communication delays, and psychological stressors demand robust analytical frameworks capable of capturing the intricate interplay between environmental conditions, individual personality traits, and collective behaviors. Among the available computational approaches, ABM has become a practical tool for examining how team processes unfold under spaceflight conditions. Rather than claiming to replace operational planning, these models offer structured experiments that reveal how differences in individual traits, coping styles, and interaction rules can accumulate into patterns at the team level. The value of this work lies in providing a systematic evidence base complementary to analog studies that helps mission planners and researchers anticipate potential risks and evaluate strategies to support crews. In this sense, ABM contributes not as a stand-alone decision system but as an analytic lens that extends what can be learned from empirical studies of long-duration missions [22,48].

In the context of space exploration, computational simulations have been employed to study various mission-critical factors. Parisi et al. [49] simulated anomalies observed in ISS and Apollo missions, focusing on how communication delays influence decision-making processes in Mars-like conditions. Their results revealed that signal latency can disrupt ground-crew coordination and shift tactical responsibilities toward the crew, thereby impacting team cohesion, stress management, and mission performance.

Mohanty et al. [50] examined how habitat design affects well-being and performance by simulating different spatial configurations and internal layouts. Their work highlighted how design features such as spatial arrangement, privacy levels, and resource allocation can either mitigate or exacerbate interpersonal tensions and stress, ultimately influencing team morale and efficiency.

Similarly, Orasanu et al. [51] investigated the relationship between individual stress responses and team performance in isolated, high-risk environments. Integrating physiological and psychological stress models into their simulations allowed them to demonstrate how variability in stress tolerance shaped in part by personality traits and physiological indicators directly affects group cohesion, communication, and operational efficiency. Such findings underscore the importance of identifying stress indicators and implementing countermeasures, such as targeted training and support systems, to maintain team health and high performance levels during extended missions.

While ABM offers significant advantages in modeling complex human systems, it is not without limitations. Current simulations often focus on a limited set of variables or shorter timeframes and may lack comprehensive psychological frameworks that consider the long-term evolution of personality traits, cultural factors, and dynamic coping mechanisms. As research progresses, incorporating these additional dimensions alongside improved calibration against empirical data can enhance the predictive accuracy and reliability of ABM. Ultimately, refining these models will better prepare astronauts and mission planners to anticipate psychological and social challenges, thus supporting sustained team effectiveness during the most demanding exploration endeavors.

## 1.2 Research questions and gap

While prior research has acknowledged that personality traits, cultural factors, and psychosocial stressors shape team functioning in isolated environments, there remains a notable absence of integrated, long-term predictive frameworks that fully capture these complexities under Mars mission conditions. Existing studies often emphasize singular dimensions

such as stress, team cohesion, or personality without unifying them into a comprehensive model capable of simulating multi-year missions. Furthermore, simulations rarely account for how the interplay of heterogeneous versus homogeneous personality compositions, evolving interpersonal dynamics, and prolonged autonomy under communication delays collectively influence psychological resilience and operational performance. Addressing this gap requires a more holistic, agent-based modeling (ABM) approach that transcends conventional short-duration analogs and fragmented theoretical insights.

To achieve this objective, we pose the following research questions:

1. How can a new ABM framework be developed to predict and analyze team performance in extreme environments like Mars missions?
2. How does team heterogeneity affect team dynamics, stress levels, and performance over the course of a simulated Mars mission?
3. What are the relationships between stress and performance in heterogeneous versus homogeneous teams under prolonged mission conditions?

Our study seeks to provide insights into optimal team compositions for long-duration space missions and contribute to the development of strategies to enhance team performance and mission success. The paper extends a preliminary version of this research [52], introducing new findings and offering a comprehensive analysis of the personality types under examination. This expanded approach facilitates a deeper understanding of the effects of personality diversity on team performance.

This paper is structured as follows: Sect 2 outlines the methodology, including the conceptual framework and the ABM approach used to simulate team dynamics in long-duration Mars missions. Sect 3 presents the simulation results, providing a comparative analysis of heterogeneous and homogeneous teams across different personality trait scenarios. In Sect 4, we discuss the implications of our findings for team composition and performance in space missions. Finally, Sect 5 summarizes the conclusions of the study and suggests avenues for future research to enhance team performance in long-duration space exploration.

## 2 Methodology

To explore these questions, we develop a new ABM framework that integrates psychological theories of stress and performance with team composition variables. This framework allows us to simulate and analyze how heterogeneous and homogeneous teams perform during long-duration Mars missions. By using this approach, we gain insights into how different team structures influence team dynamics and mission outcomes under the unique stressors of Mars expeditions.

### 2.1 Conceptual framework

Fig 1 presents the simulation's conceptual framework, structured according to the Input–Mediator–Output–Input (IMOI) model proposed by Ilgen et al. [53]. This framework extends the traditional Input–Mediator–Output (IMO) model by emphasizing the recursive nature of team dynamics. In the context of long-duration space missions, team outputs such as cohesion, performance, and health do not represent endpoints but instead feed back into the system, influencing subsequent inputs and mediators.

The simulation starts with inputs related to team and individual attributes, such as personality traits based on the FFM, skills, crew size, initial cohesion, and cultural adaptation, establishing initial conditions and shaping agent interactions. Personality traits influence

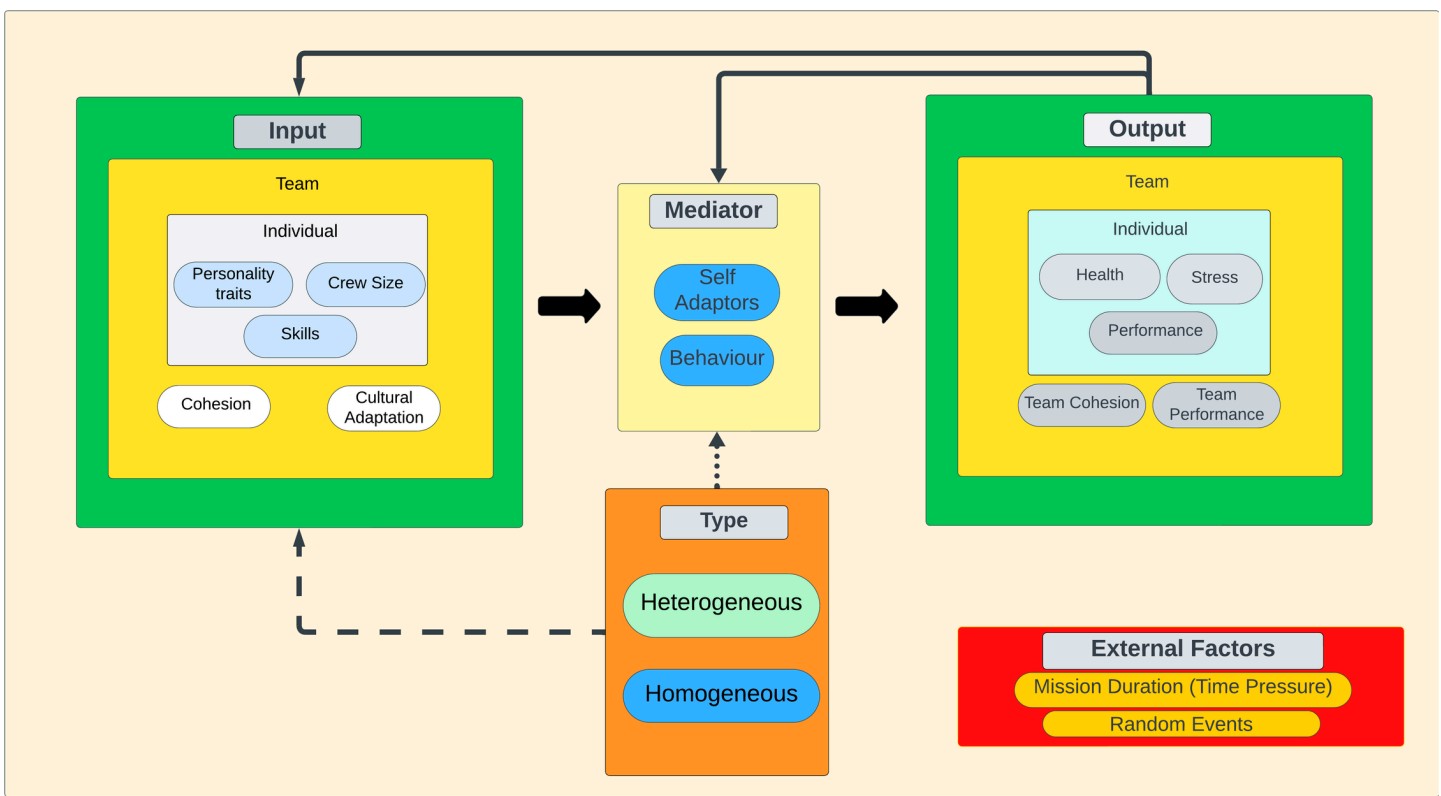

**Fig 1. Input-Mediator-Output-Input (IMOI) - ABM.**

perceptions, stress responses, and interaction styles, while skills determine task performance capabilities. Mediating processes involve behavioral adaptations and self-regulation mechanisms that agents employ to interact with each other and the environment, affecting stress management, adaptation to challenges, and collaboration. Agents utilize coping strategies, communication patterns, and conflict resolution behaviors, modulated by their personality traits and experiences.

Outputs include metrics like individual health status, stress levels, performance efficiency, and team cohesion, reflecting mission success under long-duration stressors. Individual performance depends on stress and health, while team performance emerges from collective efficiency and cohesion. External factors, such as mission duration (time pressure) and random mission events, introduce variability and affect performance, requiring agents to adapt over time and simulating the unpredictable nature of space missions. Incorporating events like equipment malfunctions, communication delays, or environmental hazards enhances realism.

Simulations were conducted for two team types: heterogeneous and homogeneous. Heterogeneous teams consist of agents with diverse personality traits and backgrounds, promoting varied perspectives and problem-solving approaches. Homogeneous teams comprise agents with similar personality profiles and cultural backgrounds, leading to uniform behaviors and interactions. Team composition influences stress coping, collaboration, and adaptation throughout the mission. Feedback loops from outputs to inputs allow outcomes such as stress levels and performance to impact future behaviors and conditions, enabling continuous

adaptation and capturing the evolving nature of team dynamics in response to accumulated experiences. Increased stress may lead to decreased performance and health deterioration, affecting team cohesion and individual coping strategies.

This comprehensive model of the Mars mission environment enables examination of how personality diversity affects team dynamics and individual responses under extreme conditions, providing insights into strategies for optimizing team performance and resilience in long-duration space exploration.

## 2.2 Model framework

The ABM is implemented using the Mesa framework in Python, an open-source library designed for developing agent-based models [54]. Mesa provides tools for agent scheduling, data collection, and visualization, facilitating the development and analysis of the simulation. The model simulates a crew of astronaut agents over a 500-day mission duration, capturing daily interactions, task performance, stress levels, and overall team dynamics.

## 2.3 Team composition conditions

As noted in the previous sections, both personality and role diversity have been highlighted in spaceflight and analog research as important determinants of team functioning [6,13,14]. Building on this recognition, our study formalizes these two dimensions in a controlled simulation setting. We adopt a 2 × 2 factorial design that systematically varies personality heterogeneity and skill/role heterogeneity, thereby enabling a structured comparison of their independent and joint effects on stress, cohesion, and performance. In doing so, the model bridges two perspectives often treated separately: the psychological lens that emphasizes trait-driven responses under stress, and the operational lens that stresses distribution of expertise across a small crew.

Personality heterogeneity is operationalized as the degree of variance in astronaut agents' Big Five profiles (low vs. high), reflecting either homogeneous or diverse psychological dispositions. Skill heterogeneity is defined by whether the crew is homogeneous in operational roles or differentiated into complementary functions. Together these dimensions yield four team composition conditions, summarized in Table 1. By structuring the design in this way, the simulation can isolate not only main effects but also potential interaction effects between psychological and functional diversity, which are difficult to study directly in spaceflight or analog environments.

Table 1. 2 × 2 Factorial design: Personality and skill/role heterogeneity.

| Condition | Low Skill Heterogeneity | High Skill Heterogeneity |
|---|---|---|
| **Low Personality Heterogeneity** (SD ≈ 0.05) | **A – Homo–Skill + Personality**<br>Agents share nearly identical Big Five profiles.<br>*Roles:* Same role assigned across crew (e.g., all engineers).<br>*Proficiency:* Uniform. | **B – Hetero–Skill**<br>Agents share similar Big Five profiles.<br>*Roles:* Crew differentiated into distinct functions (e.g., engineer, medic, pilot).<br>*Proficiency:* Varies by domain. |
| **High Personality Heterogeneity** (SD ≈ 0.20) | **C – Hetero–Personality**<br>Agents differ in Big Five profiles.<br>*Roles:* Same role assigned across crew.<br>*Proficiency:* Uniform. | **D – Hetero–Skill + Personality**<br>Agents differ in Big Five profiles.<br>*Roles:* Crew differentiated into distinct functions.<br>*Proficiency:* Varies by domain. |

**Note:** The 2 × 2 design crosses personality heterogeneity (low vs. high) with skill/role heterogeneity (low vs. high). For consistency across figures, tables, and discussion, we use the shorthand (A) Homo–Skill + Personality, (B) Hetero–Skill, (C) Hetero–Personality, and (D) Hetero–Skill + Personality.

In low skill heterogeneity conditions, all agents perform the same role with identical proficiencies, simulating a functionally uniform crew where redundancy is high but specialization is absent. In high skill heterogeneity conditions, agents are distributed across complementary roles such as pilot, engineer, medic, and scientist each with domain specific strengths. Although the model does not explicitly encode interdependence, these role differences influence how agents contribute to mission objectives and shape the distribution of workload. For instance, a medic responds more effectively to health-related events, while an engineer performs better in technical maintenance scenarios. This structure allows us to evaluate whether variability in stress responses, health outcomes, and performance metrics arises primarily from personality composition, from role allocation, or from the interaction between the two dimensions.

## 2.4 Agent initialization

Each astronaut agent is initialized with attributes that reflect both individual and team-level characteristics. These attributes include personality traits based on the FFM, skills and roles within the crew, initial stress and health levels, and social parameters. Table 2 summarizes the initialization parameters for agents in homogeneous and heterogeneous teams.

The crew size of six astronauts aligns with mission planning considerations to balance operational needs and resource constraints [55,56]. Homogeneous teams are characterized by agents with minimal variability in personality traits (standard deviation [SD] = 0.05), representing uniformity that may enhance cohesion but may limit diverse problem-solving perspectives. In contrast, heterogeneous teams include agents with greater variability (SD up to 0.2) in personality traits, reflecting diversity that can foster creativity and adaptability but may introduce potential conflicts [9].

Personality traits are assigned based on studies of astronaut profiles, with astronauts typically scoring high on conscientiousness and agreeableness, and low on neuroticism [27]. For homogeneous teams, agents have trait values clustered closely around the mean with low variability, while in heterogeneous teams, agents have a wider distribution of trait values to reflect greater diversity.

Table 2. Initialization parameters for agent attributes in homogeneous and heterogeneous teams.

| Attribute | Homogeneous Team | Heterogeneous Team | References |
|---|---|---|---|
| **Crew Size** | 6 astronauts | 6 astronauts | [55,56] |
| **Personality Traits** | | | [57] |
| Openness (O) | Mean = 0.5, SD = 0.05 | Mean = 0.5, SD = 0.2 | |
| Conscientiousness (C) | Mean = 0.8, SD = 0.05 | Mean = 0.8, SD = 0.1 | |
| Extraversion (E) | Mean = 0.5, SD = 0.05 | Mean = 0.5, SD = 0.2 | |
| Agreeableness (A) | Mean = 0.7, SD = 0.05 | Mean = 0.7, SD = 0.1 | |
| Neuroticism (N) | Mean = 0.3, SD = 0.05 | Mean = 0.3, SD = 0.1 | |
| **Skills and Roles** | Overlapping skill sets | Diverse skill sets | [6] |
| **Task Proficiency** | Equal proficiency | Varied proficiency | [6] |
| **Initial Stress Level** | 0.2 (Scale 0–1) | 0.2 (Scale 0–1) | [58] |
| **Initial Performance Level** | 1.0 (Optimal) | 1.0 (Optimal) | [32] |
| **Social Support Level** | 0.5 (Neutral) | 0.5 (Neutral) | [59] |
| **Initial Health Status** | 0.9 (Optimal) | 0.9 (Optimal) | [13] |
| **Team Cohesion** | 0.7 (High) | 0.7 (High) | [7] |

All values are normalized.

Skills are assigned to promote redundancy and interchangeability in homogeneous teams [7], ensuring that all members have overlapping skill sets. In heterogeneous teams, skills are more specialized, promoting diversity in expertise and perspectives [6].

Initial stress levels are set low for all agents (0.2 on a scale of 0–1) due to the extensive training in stress management that astronauts undergo prior to missions [58]. Initial health status is set high (0.9), reflecting optimal physical and mental preparation [13]. Social support levels are neutral (0.5), indicating that initial social relationships among crew members are neither strongly positive nor negative [5].

Team cohesion is initially set high (0.7), consistent with the expectation that astronauts begin missions with strong team unity [7].

To further investigate the impact of personality composition on team dynamics, different scenarios were simulated by varying the distributions of personality traits among agents. These scenarios are detailed in the Results section for sensitivity analysis, where their effects on team performance are analyzed.

**2.4.1 Modeling stress and performance.** Agents' stress and performance levels are dynamically updated at each simulation step based on individual factors, team interactions, and mission events. The modeling approach integrates psychological theories and empirical findings relevant to long-duration missions. Understanding the stress-performance relationship is crucial for comprehending how astronauts cope with mission demands.

Stress is modeled as a cumulative measure, influenced by various factors that represent distinct sources of stress for each agent. These components include task-related stress, social interactions, personality traits, environmental conditions, and fatigue due to workload and sleep deprivation.

The **Personality Factor** ($PF_i$) accounts for individual susceptibility to stress, influenced by neuroticism ($N_i$) and conscientiousness $C_i$. Agents with higher neuroticism experience greater stress, while higher conscientiousness mitigates stress. The personality factor is given by:

$$PF_i = N_i - 0.5 \times C_i \tag{1}$$

This factor serves as a baseline modifier for the overall stress level, reflecting the inherent psychological resilience or vulnerability of each agent [57].

**Event stress** ($ES_i$) captures stress due to task demands. It is calculated as the ratio of the number of tasks assigned to an agent ($NT_i$) to the average number of tasks across all agents ($\overline{NT}$):

$$ES_i = \frac{NT_i}{\overline{NT}} \tag{2}$$

This metric reflects the cognitive and workload pressures experienced by agents as they complete mission-critical tasks [60,61].

**Time pressure** ($TP_i$) represents stress caused by time constraints. It is defined as the ratio of the time required for tasks assigned to an agent ($T_{\text{required},i}$) to the total available working time ($T_{\text{available}}$):

$$TP_i = \frac{T_{\text{required},i}}{T_{\text{available}}} \tag{3}$$

Higher time pressure indicates a lack of sufficient time to complete tasks, leading to increased stress [62].

**Social stress** ($SS_i$) results from social interactions within the team. This stress is influenced by conflict levels ($CL_i$), social support units ($SSU_i$), and agreeableness ($A_i$). The equation for social stress is:

$$SS_i = \left( \frac{CL_i}{CL_{max}} \cdot (1 - A_i) \right) - \left( \frac{SSU_i}{SSU_{max}} \cdot A_i \right) \tag{4}$$

Agents with higher agreeableness and greater social support experience reduced social stress, while conflict increases stress levels [63,64,70].

**Environmental Stress** ($EnvS$) reflects stress induced by the environmental conditions of the mission, such as habitat constraints and external threats. It is modeled as a constant value with minor fluctuations, representing relatively stable yet persistent environmental challenges:

$$EnvS = \text{Constant Value} \tag{5}$$

This reflects the assumption that stress from environmental factors, such as confinement, isolation, and monotony, remains relatively stable over time with minor fluctuations. Research has shown that in environments like space habitats or Antarctic stations, individuals often adapt to persistent environmental stressors, leading to a consistent level of stress [4].

Agents' sleep hours, represented as **Sleep Hours** ($SH_i$), are adjusted dynamically based on stress levels. Higher stress leads to reduced sleep quality and duration. The equation for sleep hours ensures that sleep does not fall below a minimum operational limit of 4 hours:

$$SH_i = \max\left(4, 8 - 2 \times (S_i - 0.2)\right) \tag{6}$$

This adjustment reflects the impact of stress on agents' ability to recover from fatigue [71,72].

**Effective Fatigue** ($EF_i$) is derived from sleep deprivation and cumulative workload. It is calculated as:

$$EF_i = \frac{(8 - SH_i)}{8} + \frac{CW_i}{CW_{max}} \tag{7}$$

where ($CW_i$) represents the cumulative workload of the agent. Effective fatigue quantifies the physical and cognitive exhaustion resulting from sustained work under stress [60,66].

The **Overall Stress** ($S_i$) level combines all these factors, weighted according to their relative importance. The equation is:

$$\begin{aligned} S_i = w_{ES}ES_i + w_{TP}TP_i + w_{EF}EF_i \\ + w_{SS}SS_i + w_{EnvS}EnvS + PF_i \end{aligned} \tag{8}$$

The weights ($w_{ES}, w_{TP}, w_{EF}, w_{SS}, w_{EnvS}$) sum to 1, ensuring a balanced contribution of stress components. The overall stress level is clamped between 0 and 1 to maintain validity [60,61].

**Performance** ($P_i$) is modeled as a function of stress, following the Yerkes-Dodson law. Performance peaks at an optimal stress level ($S_{opt}$) and decreases as stress deviates from this point:

$$P_i = \exp\left(-k\left(S_i - S_{opt}\right)^2\right) \tag{9}$$

Here, ($k$) determines the sensitivity of performance to stress deviations, and the relationship captures the trade-off between stress and efficiency [67,68].

Team dynamics are also influenced by **Team Cohesion** ($TC$), which is derived from the balance between average social support units ($\overline{SSU}$) and conflict levels ($\overline{CL}$) within the team:

$$TC = \max\left(0, \min\left(1, \overline{SSU} - \overline{CL}\right)\right) \qquad (10)$$

Team cohesion affects overall team resilience and performance, ensuring that the agents operate cohesively despite stressors [9,69].

Fig 2 illustrates the stress-performance modeling framework, showing how individual, environmental, and social factors collectively influence stress and performance outcomes. Table 3 summarizes the Eqs. and their descriptions, providing an overview of the modeling approach used in the simulation.

## 2.5 Agent interactions and team dynamics

Agent interactions are central to the simulation, as they significantly influence stress levels, health status, performance, cultural adaptation, and team cohesion. These interactions simulate the confined and isolated nature of Mars missions, where interpersonal dynamics are critical to mission success.

Each day, agents interact with two others, selected randomly without replacement. These interactions represent the limited but impactful social exchanges typical in small, isolated crews [17]. Positive interactions enhance social support and reduce conflict, while negative interactions increase conflict and stress. The outcomes of these interactions are shaped by the agents' individual characteristics, such as agreeableness and neuroticism. Agents with higher

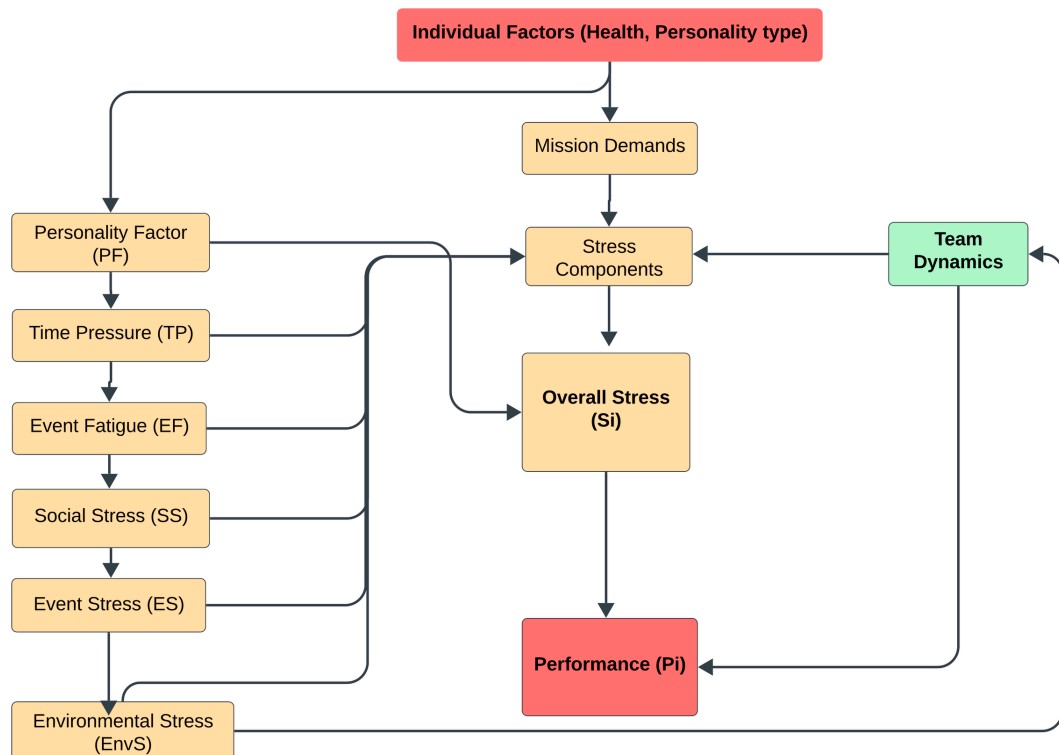

**Fig 2. Stress-performance modeling framework.**

**Table 3. Stress and performance modeling equations.**

| Metric/Assumption | Formula | Description | Ref. |
|---|---|---|---|
| **Personality Factor (PF)** | $PF_i = N_i - 0.5 \cdot C_i$ | Individual susceptibility to stress; $N_i$ is Neuroticism, $C_i$ is Conscientiousness | [57] |
| **Event Stress (ES)** | $ES_i = \dfrac{NT_i}{\overline{NT}}$ | Stress from task demands; $NT_i$ is the number of tasks for agent $i$, $\overline{NT}$ is the average number of tasks | [60,61] |
| **Time Pressure (TP)** | $TP_i = \dfrac{T_{\text{required},i}}{T_{\text{available}}}$ | Stress due to time constraints; $T_{\text{required},i}$ is time required for tasks, $T_{\text{available}}$ is available working time | [62] |
| **Social Stress (SS)** | $SS_i = \left( \dfrac{CL_i}{CL_{\max}} \cdot (1 - A_i) \right) - \left( \dfrac{SSU_i}{SSU_{\max}} \cdot A_i \right)$ | Stress from social interactions; $CL_i$ is conflict level, $SSU_i$ is social support units, $A_i$ is Agreeableness | [63,64] |
| **Environmental Stress (EnvS)** | $EnvS = $ Constant Value | Stress from environmental factors, assumed constant with minor fluctuations | [4] |
| **Sleep Hours ($SH_i$)** | $SH_i = \max\left(4, 8 - 2 \cdot (S_i - 0.2)\right)$ | Sleep adjustment based on stress; minimum of 4 hours | [58,65] |
| **Effective Fatigue (EF)** | $EF_i = \dfrac{(8 - SH_i)}{8} + \dfrac{CW_i}{CW_{\max}}$ | Fatigue from sleep deprivation and workload; $SH_i$ is sleep hours, $CW_i$ is cumulative workload | [60,66] |
| **Overall Stress ($S_i$)** | $S_i = w_{\text{ES}} \cdot ES_i + w_{\text{TP}} \cdot TP_i + w_{\text{EF}} \cdot EF_i + w_{\text{SS}} \cdot SS_i + w_{\text{EnvS}} \cdot EnvS + PF_i$ | Total stress level for agent $i$; weights sum to 1, $PF_i$ is personality factor | [61] |
| **Performance ($P_i$)** | $P_i = \exp\left(-k \cdot \left(S_i - S_{\text{opt}}\right)^2\right)$ | Performance level based on stress; $k$ determines sensitivity, $S_{\text{opt}}$ is optimal stress level | [67,68] |
| **Team Cohesion (TC)** | $TC = \max\left(0, \min\left(1, \overline{SSU} - \overline{CL}\right)\right)$ | Overall team cohesion; $\overline{SSU}$ is average social support units, $\overline{CL}$ is average conflict level | [9,69] |

agreeableness tend to foster positive interactions and resolve conflicts effectively [70], while those with higher neuroticism are more likely to experience negative interactions, leading to elevated stress and conflict.

Team cohesion emerges as a result of the balance between social support and conflict within the group. Higher levels of social support enhance cohesion, while increased conflict reduces it [9]. This dynamic captures the evolving interpersonal relationships in a high-stress, confined environment and their collective impact on team performance.

To introduce the unpredictability of real missions, the model incorporates stochastic mission events, such as equipment failures, communication delays, or environmental hazards. These events add to the environmental stress experienced by agents and require adaptive responses, reflecting the challenges documented in long-duration space missions [7].

Cultural adaptation plays a significant role in heterogeneous teams composed of individuals from diverse backgrounds. Agents with higher openness are better able to adapt to cultural differences, reducing stress and fostering stronger team cohesion [5]. These dynamics emphasize the potential advantages and challenges of team diversity in extreme environments.

The simulation framework integrates principles from environmental psychology [4], emphasizing the subjective experiences of agents based on their personality traits, interpersonal interactions, and mission events. The approach ensures that the model reflects realistic variations in how individuals perceive and respond to similar circumstances. By analyzing

both homogeneous and heterogeneous teams, the framework enables a comparative analysis of how team composition affects stress, performance, and cohesion.

Assumptions in the model are grounded in empirical data from space missions and analog environments. Personality traits, stressors, and interaction outcomes are modeled based on validated findings from team dynamics and psychological studies [5,17].

The initial parameters, including the crew size of six astronauts, mission duration, and starting values for stress, health, and performance, are based on realistic mission planning considerations and astronaut selection criteria [56]. These inputs ensure that the model reflects the operational and psychological constraints of Mars missions, providing a basis for studying team dynamics in prolonged isolation and extreme environments.

## 3 Results

This section presents a comparative analysis of the outcomes for homogeneous and heterogeneous teams across three personality trait scenarios over the duration of a 500-day Mars mission. The scenarios examined are as follows:

1. High Conscientiousness and Low Neuroticism
2. Balanced Personality
3. High Extraversion and High Agreeableness

Table 4 summarizes these scenarios and their associated trait.

In addition to personality profiles, we varied astronaut skill set distributions to reflect homogeneous and heterogeneous role configurations, as summarized in Table 5.

In combination, the personality scenarios and skill configurations create a 2 × 2 factorial design with four team composition conditions: (A) Homo–Skill + Personality, (B) Hetero–Skill, (C) Hetero–Personality, and (D) Hetero–Skill + Personality. These categories enable a controlled comparison of how psychological and functional diversity shape team dynamics under varying trait configurations. The following results are organized to highlight trends across these four team types under each personality scenario, focusing on key mission relevant outcomes such as stress, performance, health, and cohesion.

Table 4. Personality trait scenarios for astronaut teams.

| Scenario | Description and Trait Ranges | References |
| --- | --- | --- |
| Scenario 1: High Conscientiousness & Low Neuroticism | Astronauts with high conscientiousness (0.8–0.9) and low neuroticism (0.1–0.3) are reliable, organized, and less prone to stress. Traits: Openness (0.6–0.7), Agreeableness (0.5–0.7), Extraversion (0.5–0.7). | [31] |
| Scenario 2: Balanced Personality | Astronauts with average levels across Big Five traits serve as a baseline. Traits: Conscientiousness (0.4–0.6), Neuroticism (0.4–0.6), Openness (0.4–0.6), Agreeableness (0.4–0.6), Extraversion (0.4–0.6). | [33] |
| Scenario 3: High Extraversion & High Agreeableness | Teams with high extraversion (0.8–0.9) and agreeableness (0.8–0.9) may enhance team cohesion but face focus challenges. Traits: Conscientiousness (0.4–0.6), Neuroticism (0.4–0.6), Openness (0.5–0.7). | [27] |

Table notes summarize personality trait scenarios for heterogeneous and homogeneous astronaut teams during the simulation.

**Table 5**. Functional skillset profiles for astronaut teams.

| Team Type | Description and Skill Vectors (Engineer, Medic, Scientist, Pilot) | References |
|---|---|---|
| **Homogeneous Skills (Low Skill Diversity)** | All astronauts share a common functional role, with similar proficiency across skill domains. *Skill Vector per Agent:* *e.g., Engineer: [0.85, 0.30, 0.40, 0.30]* | [13] |
| **Heterogeneous Skills (High Skill Diversity)** | Each astronaut is assigned a distinct mission-critical role, resulting in diverse skill profiles. *Skill Vectors:* (Engineer): [0.90, 0.20, 0.40, 0.35] (Medic): [0.30, 0.85, 0.35, 0.30] (Scientist): [0.40, 0.30, 0.90, 0.35] (Pilot): [0.30, 0.30, 0.30, 0.90] | [22,33] |

Skill values are normalized between 0 and 1 to represent agent proficiency in each operational domain. Homogeneous teams simulate generalist roles, while heterogeneous teams reflect specialized assignments per astronaut.

## 3.1 Average metrics across scenarios

**3.1.1 Homogeneous teams.** Homo–Skill + Personality case is composed of members with similar personality traits and functional skills, the team exhibited the lowest overall outcomes among all tested configurations. As shown in Fig 3, final stress levels reached 0.96, the highest observed in this scenario, while health dropped to 0.39 and performance remained low at 0.39. The uniformity in both psychological and functional profiles may have limited the team's ability to respond flexibly to prolonged stress exposure. With fewer differences in coping strategies or task approaches, the team might have struggled to adapt as challenges accumulated over time.

**3.1.2 Time-based metrics – Homogeneous team.** Fig 4 presents the average trajectories of stress, performance, and health over the 500-day mission for homogeneous teams

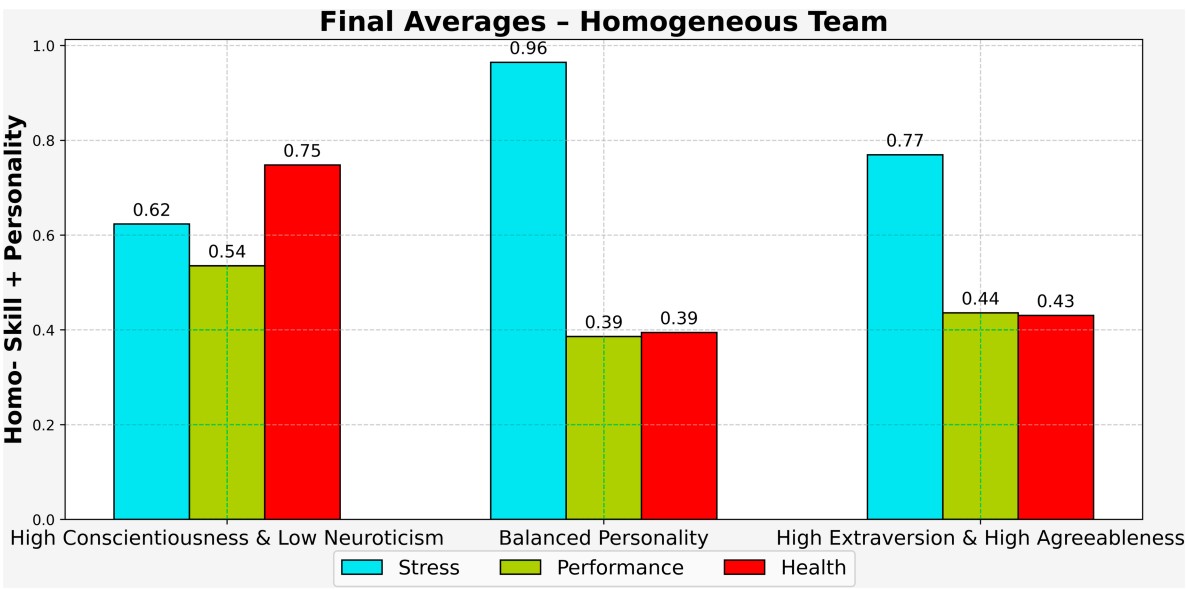

**Fig 3. Final Avg. for Homo-Team (500 Days).**

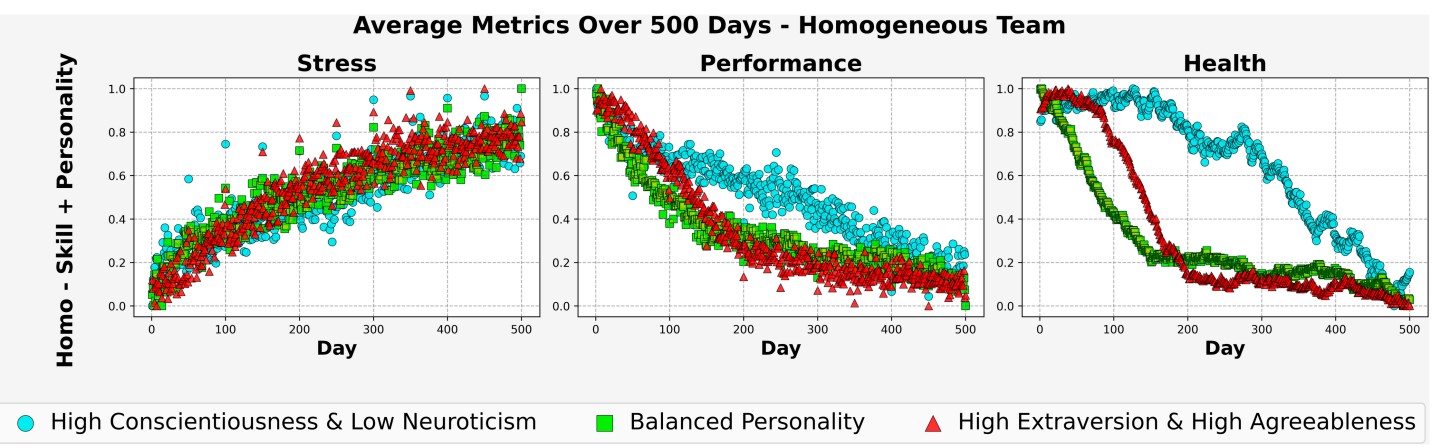

**Fig 4. Avg. Metrics for Homo-Team (500 days).**

(*Homo–Skill + Personality*) across three personality trait profiles. In the *Balanced Personality* condition, stress escalated rapidly and leveled off near (0.85). Performance declined sharply over time, stabilizing around (0.2), while health dropped below (0.1) by the final phase. This uniform configuration may also have lacked the psychological diversity needed to buffer cumulative stress, contributing to accelerated declines in both functioning and well-being.

In contrast, the *High Conscientiousness and Low Neuroticism* condition showed a slower and lower rise in stress, remaining near (0.65) toward the end. Performance declined gradually and stayed above (0.4) for most of the mission. Health remained high during the first three-quarters of the timeline and only dropped significantly after day 400, ending just above (0.2), suggesting this trait profile may support greater long-term stability and adaptive functioning under sustained isolation.

The *High Extraversion and High Agreeableness* team showed an early rise in stress, plateauing near (0.75). Performance declined at a rate similar to the Balanced group but with slightly higher values toward the end, settling near (0.25). Health deteriorated steadily after day 100 and approached the lower range by mission completion.

Taken together, these patterns suggest that homogeneous teams composed of individuals with Balanced Personality traits may be most vulnerable to performance and health degradation, while teams with High Conscientiousness and Low Neuroticism may better withstand the psychological demands of long-duration missions.

**3.1.3 Heterogeneous teams.** In heterogeneous teams, outcomes varied depending on the combination of personality traits and skill diversity applied under each scenario.

Fig 5 presents the final average values for stress, performance, and health across the three heterogeneous configurations. In the *High Conscientiousness and Low Neuroticism* scenario, the hetero-skill + personality team achieved the most favorable results, with the lowest stress (0.58), highest performance (0.58), and highest health (0.78). The hetero-personality team showed moderate stress (0.71), performance (0.52), and health (0.74). The hetero-skill team had the highest stress (0.79), and lower performance (0.44) and health (0.47).

Under the *Balanced Personality* scenario, all heterogeneous teams showed elevated stress and reduced performance and health. The hetero-skill + personality team had stress at 0.82, with performance and health at 0.43 and 0.47, respectively. The hetero-skill team had similar stress (0.79), identical performance (0.44), and health (0.47). The hetero-personality team had the highest stress (0.85), with slightly lower performance (0.42) and health (0.45).

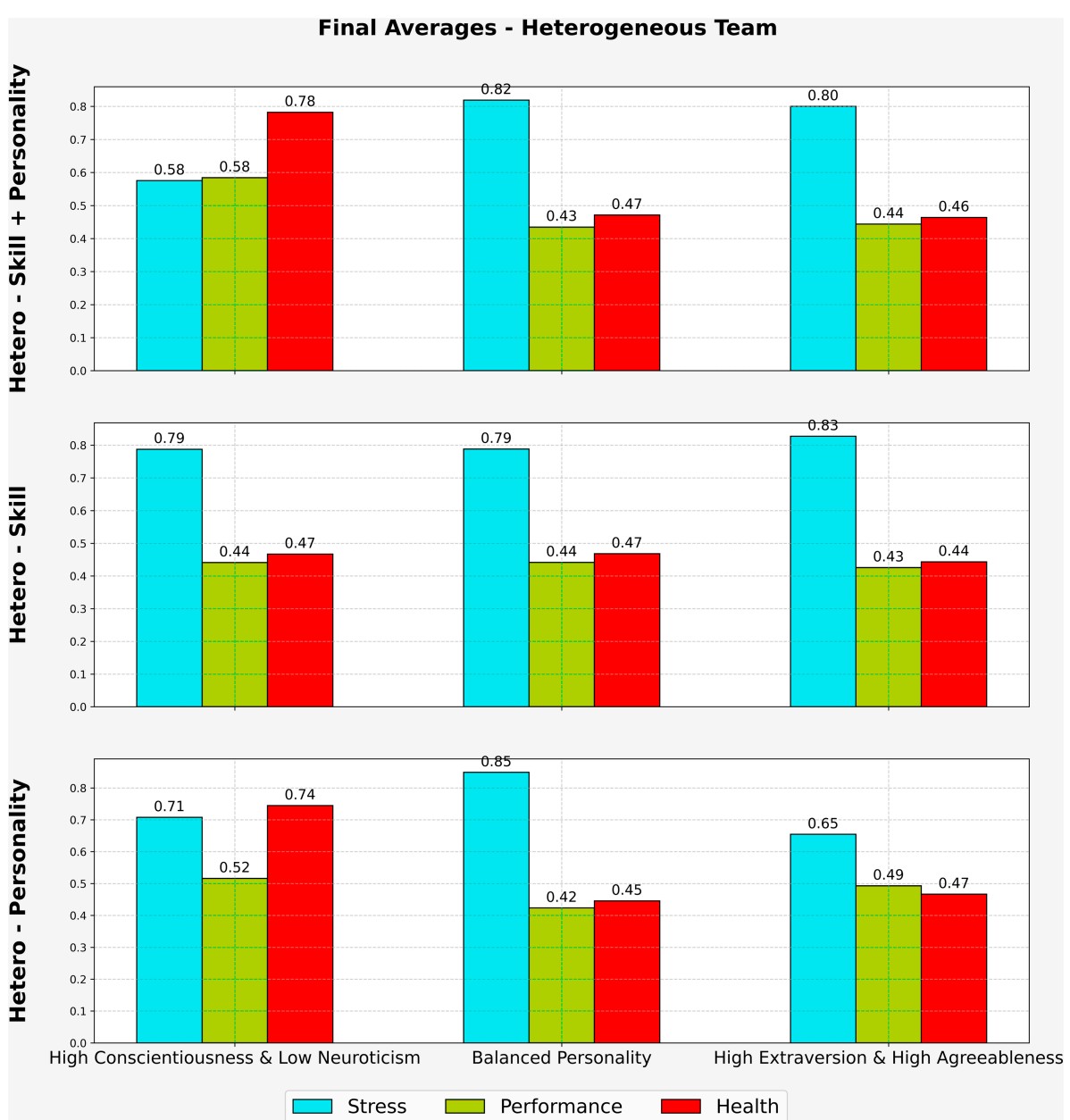

**Fig 5. Final Avg. for Hetero Team (3 Cases, 500 days).**

In the *High Extraversion and High Agreeableness* scenario, the hetero-personality team showed the best overall outcomes, with the lowest stress (0.65), highest performance (0.49), and health (0.47). The hetero-skill + personality team had slightly higher stress (0.80), with performance at (0.44) and health at (0.46). The hetero-skill team exhibited the highest stress (0.83), with performance and health at (0.43) and (0.44), respectively.

These findings suggest that the hetero-skill + personality configuration may offer more favorable outcomes in high-conscientiousness settings and could remain comparatively resilient across different personality conditions. In contrast, hetero-skill teams alone appeared

to experience consistently elevated stress and reduced performance, potentially reflecting limited psychological adaptability.

**3.1.4 Time-based metrics – Heterogeneous team.** Fig 6 illustrates the evolution of average stress, performance, and health over the 500-day mission across three personality trait scenarios. Each row corresponds to one heterogeneous team configuration hetero-skill + personality, hetero-personality, and hetero-skill while each column represents one of the core metrics.

In the *High Conscientiousness and Low Neuroticism* scenario, the hetero-skill + personality configuration appeared to perform more favorably across the three metrics. It showed the lowest average stress (0.36), relatively high performance (0.69), and the most stable health levels (0.92). The hetero-personality team followed closely with stress around (0.43), performance near (0.63), and health approaching (0.88). In contrast, the hetero-skill team exhibited higher stress (0.56), with moderately lower performance (0.54) and health (0.62).

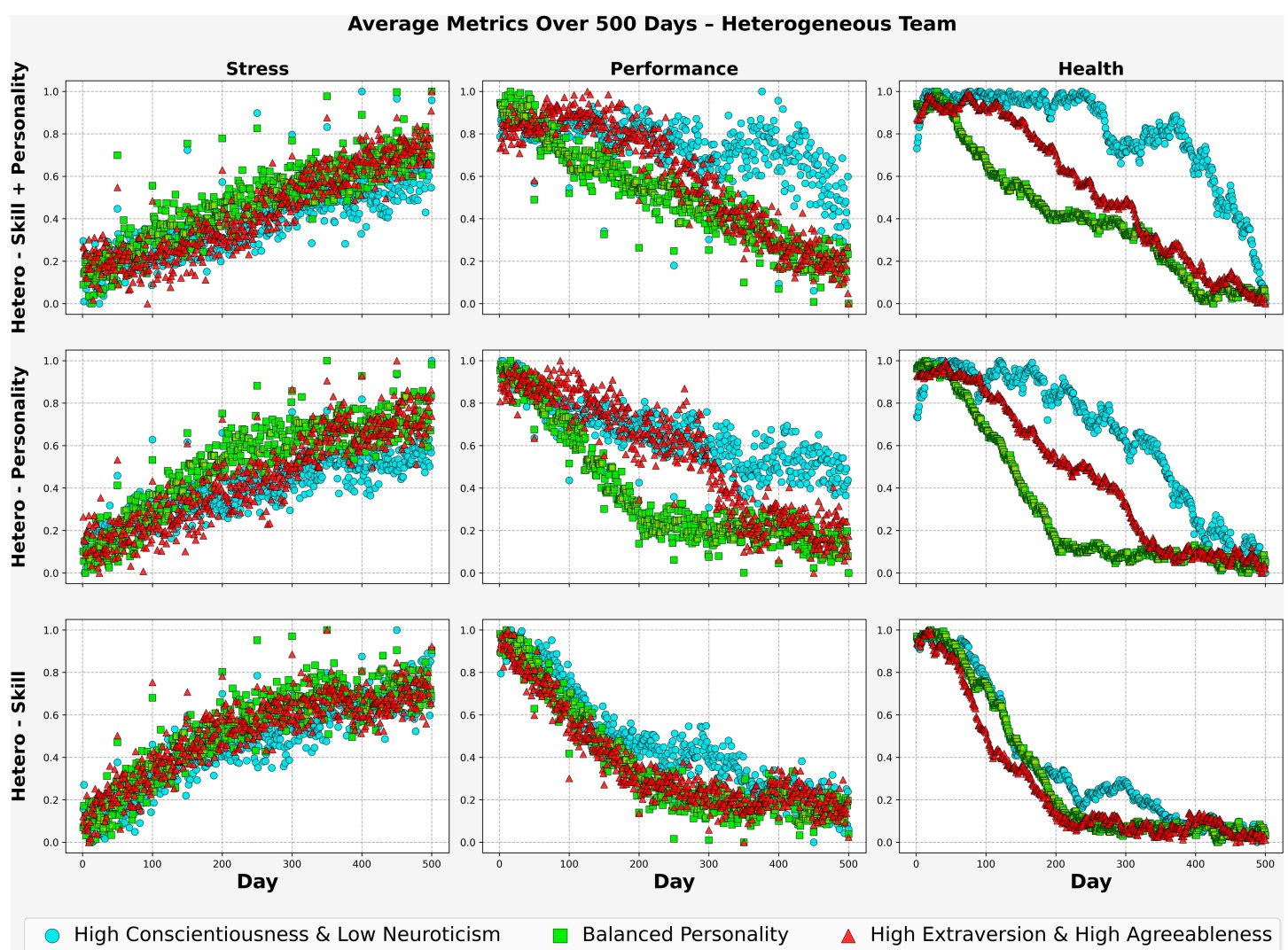

**Fig 6. Avg. Metrics for Hetero-Team (3 Cases, 500 days).**

For the *Balanced Personality* scenario, all configurations experienced more pronounced declines in health and performance, while stress increased over time. The hetero-skill + personality team showed average stress near (0.53), performance around (0.56), and health at approximately (0.64). The hetero-personality team followed with stress around (0.59), performance near (0.52), and health at (0.57). The hetero-skill configuration exhibited slightly higher stress (0.58), with performance and health declining to around (0.52) and (0.61), respectively. These patterns could reflect reduced psychological buffering effects when personality traits are more moderate.

In the *High Extraversion and High Agreeableness* condition, the hetero-personality team appeared to achieve the most favorable average outcomes: stress remained lower (0.51), while performance and health were higher (0.58) and (0.67), respectively. The hetero-skill + personality team also maintained relatively stable metrics, with stress near (0.48), performance at (0.61), and health around (0.72). The hetero-skill configuration recorded higher stress (0.59) and comparatively lower performance (0.51) and health (0.56). These results imply that under socially expressive trait profiles, personality diversity may play a more critical role in sustaining team performance and well-being than functional diversity alone.

### 3.2 Individual metrics: Balanced personality – Nominal case

For the *Balanced Personality* scenario, individual level trajectories of stress, performance, and health were tracked over the 500-day mission to compare homogeneous and heterogeneous team dynamics.

**3.2.1 Homogeneous teams.** Fig 7 presents the individual stress, performance, and health metrics for astronauts in the homogeneous team configured with similar skill and personality profiles (Homo–Skill + Personality). Stress levels increased consistently across all six astronauts, beginning near (0.35–0.45) and rising above (0.85) for most by the end of the mission. Several individuals approached near-maximal stress during the final quarter, reflecting a shared vulnerability to prolonged exposure.

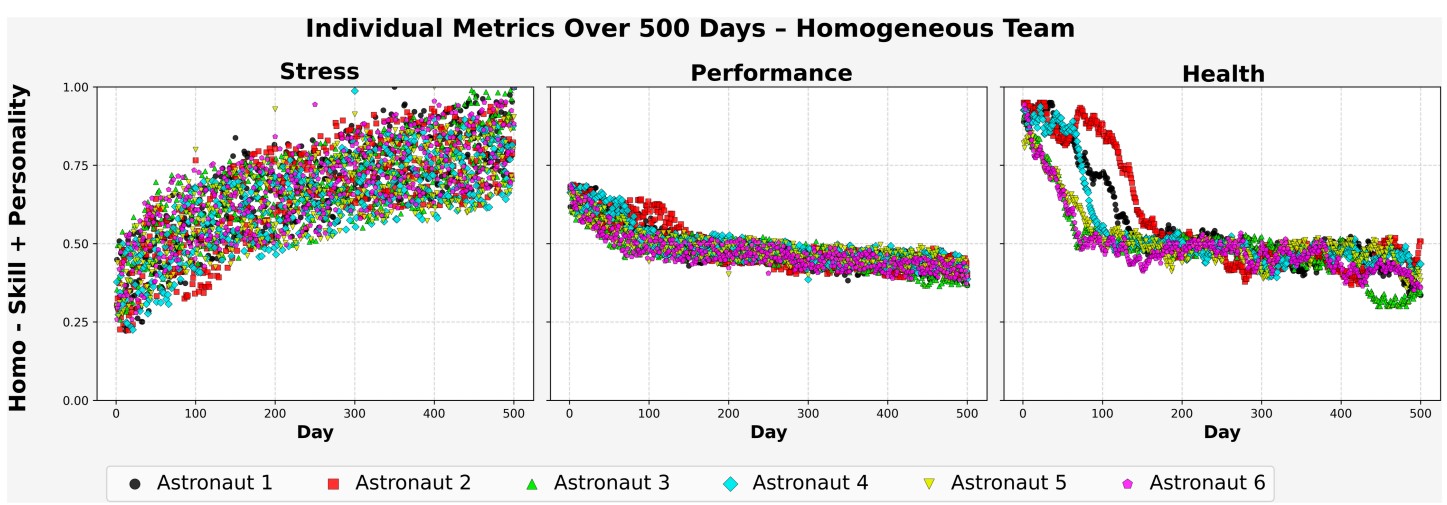

**Fig 7. Homo team – balanced personality.**

Performance followed a gradual downward trend, starting around (0.70) and declining below (0.50) after day (250). By the mission's end, all agents converged within a narrow range between (0.42) and (0.47), indicating a uniform loss in task efficiency over time.

Health deteriorated more rapidly and earlier than the other metrics. Most astronauts experienced a steep decline between days (50) and (150), with values falling below (0.40) and stabilizing between (0.25) and (0.35) for the remainder of the mission. Few signs of recovery were evident after the initial drop.

These patterns point to a narrow range of behavioral responses in the team. With shared psychological and functional profiles, the group exhibited little divergence in stress adaptation or functional resilience. The absence of heterogeneity in both skill and personality may have constrained the team's ability to distribute load or recover under stress, contributing to mission-long degradation across all agents.

**3.2.2 Heterogeneous teams.** Fig 8 displays individual stress, performance, and health metrics over the 500-day mission for heterogeneous teams configured with (1) hetero-personality, (2) hetero-skill, and (3) hetero-skill + personality.

In the *hetero-personality* condition (bottom row), stress gradually increased for all astronauts, ending between (0.70) and (0.88). Performance declined steadily across the mission, with most agents converging around (0.42–0.46) by the final phase. Health showed wider variability some astronauts experienced early declines below (0.45) after day (200), while others maintained moderate health levels near (0.50) into the final third of the mission. This spread may reflect the benefits of trait-based heterogeneity in enabling diverse coping responses.

In the *hetero-skill* configuration (middle row), stress increased at a slightly faster rate than in the personality-based case, with most agents nearing or exceeding (0.85). Performance dropped more sharply during the second half of the mission, with end values commonly between (0.46) and (0.50). Health deteriorated earlier by day (300), nearly all astronauts had dropped below (0.50).

The *hetero-skill + personality* configuration (top row) showed the most favorable balance. Stress levels rose moderately and remained more evenly distributed across individuals, typically ranging from (0.60) to (0.80) by mission end. Performance declined more gradually and ended closer to (0.48–0.52) for most astronauts. Health also followed a more stable trajectory, with several agents maintaining values near or above (0.55) for much of the mission, even as others declined after day (250).

These individual patterns reinforce the value of integrating both psychological and functional diversity. Teams structured with hetero-skill + personality appeared better able to maintain performance and withstand long-term stressors, likely due to a wider range of adaptive strategies and complementary contributions among crew members.

**3.2.3 Comparison.** Under the *Balanced Personality* condition, heterogeneous teams outperformed the homogeneous team across both average outcomes and individual level trajectories. Teams configured with hetero-personality and hetero-skill + personality exhibited comparatively lower stress, better performance retention, and more stable health throughout the mission.

The homogeneous team ended up with the highest final stress value (0.96), and the lowest performance (0.39) and health (0.39) scores. This pattern of decline was also evident in individual trajectories, where all six agents exhibited nearly identical degradation curves, particularly sharp early declines in health and converging stress levels above (0.85) by the final phase. Minimal intra-team variability suggested a lack of buffering capacity and resilience under cumulative stress.

Among the heterogeneous teams, the hetero-personality group ended with slightly lower stress (0.85), marginally better performance (0.42), and health (0.45) values. The hetero-skill

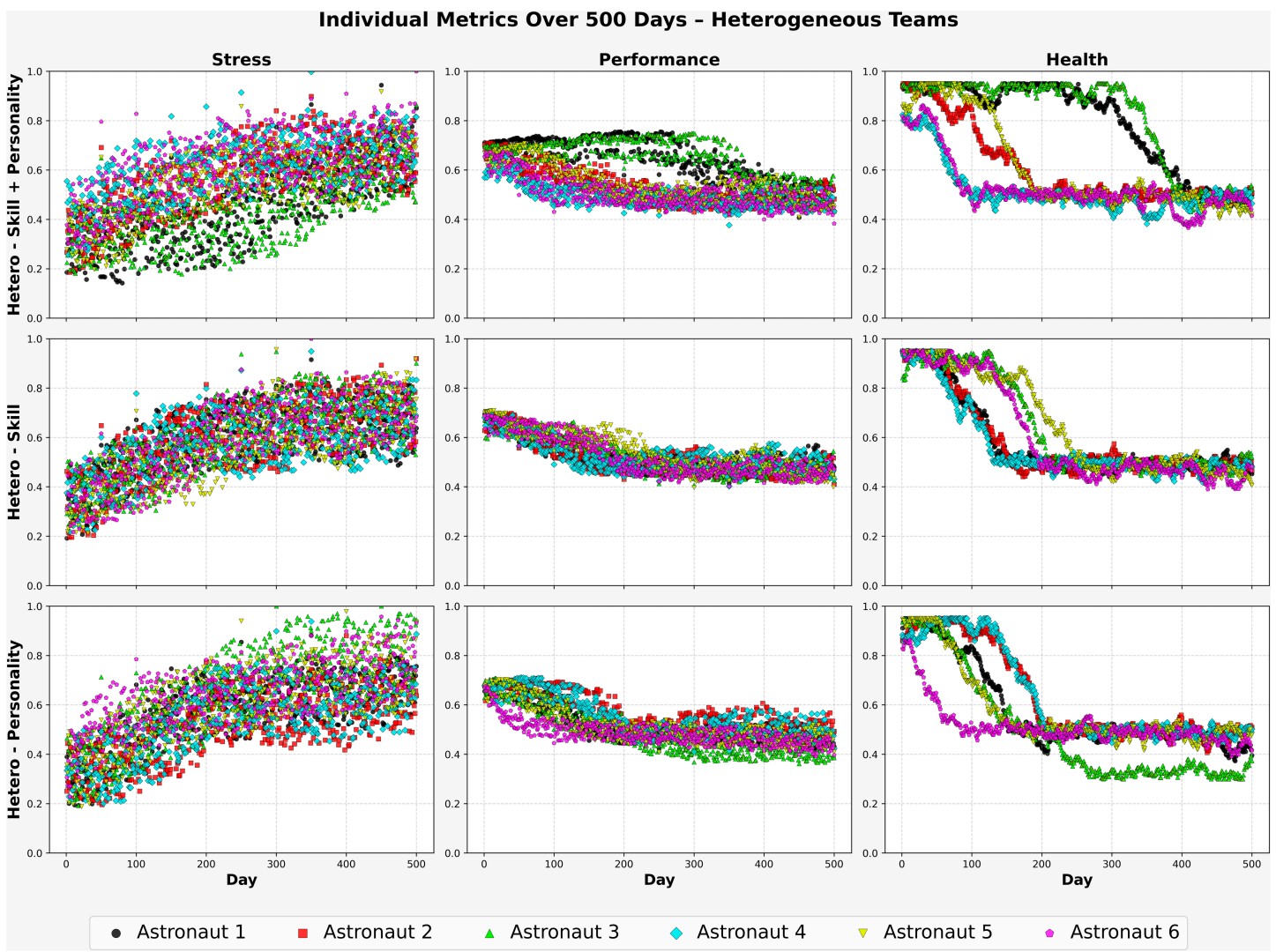

**Fig 8. Hetero team – Balanced personality (3 Cases).**

team showed similar outcomes, with stress at (0.79), performance at (0.44), and health at (0.47). The hetero-skill + personality team concluded with stress at (0.82), performance at (0.43), and health at (0.47). While the final values across heterogeneous teams were close, their individual-level patterns revealed smoother declines and greater within team variability, indicating the presence of diverse coping responses and recovery behaviors.

Individual trajectories in heterogeneous teams displayed more varied stress and health adaptations, with some astronauts preserving functional capacity longer than others. In contrast, the homogeneous team showed limited divergence across all agents, reinforcing the interpretation that shared psychological and functional traits may constrain adaptive flexibility in prolonged, high-demand environments.

Taken together, these results suggest that incorporating both psychological and functional heterogeneity, especially in combination, may offer a more robust defense against cumulative stress and performance degradation in long-duration missions. However, the magnitude of

this benefit appears contingent on the specific personality trait configurations. In particular, teams characterized by higher conscientiousness and lower neuroticism tended to experience slower stress accumulation and more resilient functional trajectories over time.

Although the simulation does not aim to forecast operational outcomes directly, it provides a structured platform to examine how specific team compositions influence group dynamics over time. These findings may contribute to the design of more adaptive crew selection strategies and inform mission planning for extended spaceflight scenarios.

### 3.3 Team cohesion in the 2 × 2 composition scenarios

Fig 9 illustrates the evolution of cohesion across four team configurations based on a 2 × 2 factorial design: *homo-skill + personality*, *hetero-skill*, *hetero-personality*, and *hetero-skill + personality*. Each subplot presents individual cohesion trajectories for six astronauts over the 500-day mission, where values range from 0 (no cohesion) to 1 (maximum cohesion).

While cohesion is conceptually a group-level construct, the model tracks each astronaut's average pairwise cohesion with the rest of the team over time. These values are not individual traits, but reflect each agent's degree of social integration within the group, based on the strength of their reciprocal social ties. This framing enables analysis of how interpersonal connections vary dynamically across agents and scenarios.

In this model, cohesion is computed as a symmetric, undirected metric representing the average strength of mutual social ties between astronaut pairs. The cohesion matrix is updated over time based on bidirectional interactions, meaning that the cohesion between astronaut

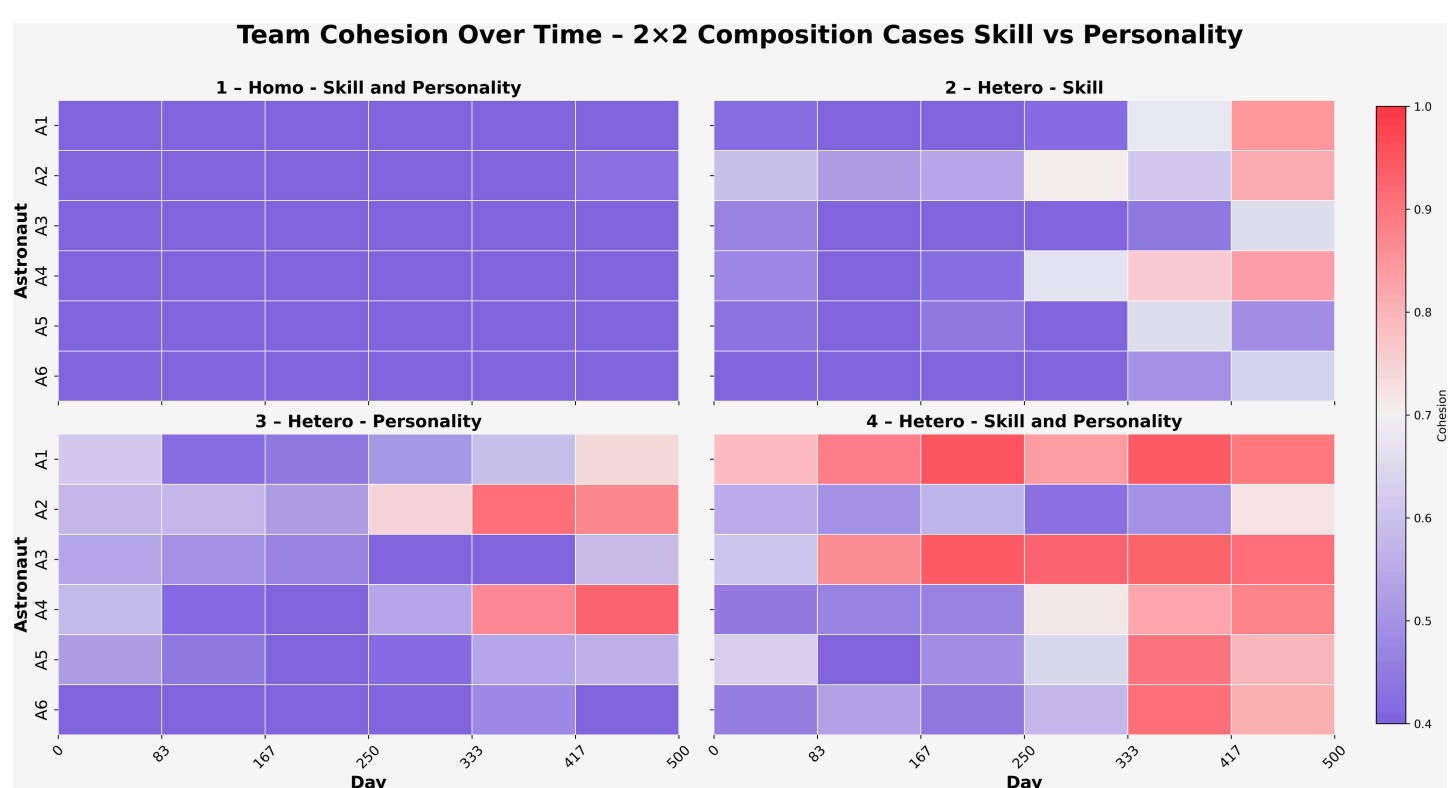

**Fig 9. Team cohesion – 2 × 2 Scenarios.**

A1 and astronaut A2 is always equal to that between A2 and A1, for example. One-sided perceptions or non-reciprocated relationships are not modeled separately. This design reflects the assumption that social integration arises from mutual experiences and sustained exchanges rather than unilateral perceptions.

In Condition 1 (*homo–skill + personality*), cohesion remained uniformly low throughout the 500-day mission across all astronauts. Most values clustered tightly between (0.45) and (0.50), indicating minimal growth in social bonding. This pattern may reflect limited psychological and functional complementarity, reducing opportunities for meaningful interpersonal integration.

In Condition 2 (*hetero–skill*), cohesion displayed a more varied trajectory. Astronauts A1, A2, and A4 reached values between (0.75) and (0.80) by the end of the mission, while others, such as A3, A5 and A6, remained closer to (0.50–0.60). These localized increases may reflect the emergence of task-driven dependencies and informal alliances within a functionally diverse but psychologically uniform team.

Condition 3 (*hetero–personality*) showed moderate-to-strong but uneven cohesion development. Astronauts A1, A2, and A4 exceeded (0.70) in the final phase, with A4 nearing (0.90). Others, like A1 stabilized at moderate levels near (0.65–0.75). This variability may reflect compatibility rooted in trait-driven interaction styles, where some individuals formed stronger dyadic bonds than others.

Condition 4 (*hetero–skill + personality*) yielded the highest and most consistently distributed cohesion values. Astronauts A1 and A3 exceeded (0.85) early in the mission, and by the final phase, most team members sustained cohesion levels above (0.80). This configuration, which integrates both psychological and functional diversity, may have fostered deeper mutual understanding, enhanced behavioral complementarity, and broader team-wide social integration.

Together, these findings suggest that cohesion development in long-duration missions is sensitive to both personality and skill composition. While skill heterogeneity alone may promote specific collaborative bonds, psychological diversity appears to enhance interpersonal attunement. The combination of both may offer the most robust foundation for maintaining high team cohesion under extended stress and isolation.

## 3.4 Monte Carlo analysis

To assess the robustness of core trends, a Monte Carlo analysis was performed with 1,000 stochastic runs across four team configurations: *homo–skill + personality*, *hetero–skill*, *hetero–personality*, and *hetero–skill + personality*. Each run introduced random variation in personality traits, initial stress conditions, and interaction patterns, mimicking the uncertainty and individual differences present in real missions.

The distributions in Fig 10 and summary statistics in Table 6 confirm general trends observed in core simulations. Teams with psychological diversity particularly *hetero–personality* and *hetero–skill + personality* performed better than the *homo–skill + personality* baseline. These configurations showed lower average stress (0.74–0.76), higher performance (0.49), and better health (0.52–0.53). Notably, *hetero–personality* slightly outperformed *hetero–skill + personality* on health (0.53 vs. 0.52), despite lacking functional diversity.

In contrast, teams with only skill heterogeneity (*hetero–skill*) showed limited improvements. Stress remained high (0.86), and health (0.43) remained closer to the homogeneous baseline (0.41), suggesting that functional diversity alone may not be sufficient to buffer cumulative stress or maintain physiological resilience. Performance in *hetero–skill* teams also remained the lowest among heterogeneous groups (0.41).

## Monte Carlo Results – Final Metrics Distribution

**Fig 10. Monte Carlo – Final Metrics (n = 1000, 4 Teams).**

**Table 6**. Monte Carlo results for the 4 Team configurations (n = 1000).

| Metric | Team Type | Mean | Standard Deviation | Range (Min–Max) |
|---|---|---|---|---|
| **Stress Level** | Homo-Skill + Personality | 0.84 | 0.04 | 0.75–0.95 |
| | Hetero-Skill + Personality | 0.76 | 0.05 | 0.61–0.90 |
| | Hetero-Personality | 0.74 | 0.05 | 0.60–0.91 |
| | Hetero-Skill | 0.86 | 0.05 | 0.70–0.96 |
| **Performance** | Homo-Skill + Personality | 0.42 | 0.02 | 0.38–0.47 |
| | Hetero-Skill + Personality | 0.49 | 0.02 | 0.43–0.54 |
| | Hetero-Personality | 0.49 | 0.03 | 0.42–0.56 |
| | Hetero-Skill | 0.41 | 0.02 | 0.38–0.48 |
| **Health Status** | Homo-Skill + Personality | 0.41 | 0.03 | 0.35–0.51 |
| | Hetero-Skill + Personality | 0.52 | 0.03 | 0.39–0.65 |
| | Hetero-Personality | 0.53 | 0.04 | 0.40–0.63 |
| | Hetero-Skill | 0.43 | 0.03 | 0.34–0.50 |

Homogeneous teams exhibited the most constrained distributions, but consistently under-performed in all three metrics. Their average stress remained elevated (0.84), while performance (0.42) and health (0.41) were the lowest overall. This team reflected a narrow but less adaptable team profile, vulnerable to accumulating strain over time.

Taken together, these findings suggest that psychological heterogeneity particularly in traits such as conscientiousness, neuroticism, and extraversion may play a more decisive role than skill diversity alone in supporting team performance and resilience. The broader distributions observed among heterogeneous teams also imply greater adaptive range, even if outcomes are less predictable. This variability may be advantageous under dynamic mission demands, where flexibility and stress regulation are critical.

## 4 Discussion

Simulation results highlight how team composition shapes stress, health, performance, and cohesion in long-duration space missions. In the scenarios examined, heterogeneous teams generally demonstrated better outcomes than homogeneous teams, indicating that hetero-personality and hetero-skill diversity may support team resilience under sustained operational demands.

In the scenario featuring high conscientiousness and low neuroticism, heterogeneous teams particularly those with hetero-personality or hetero-skill + personality maintained lower stress and higher health and performance levels than homogeneous teams. The presence of more varied emotional regulation and task engagement profiles may have supported adaptive functioning as the mission progressed. The hetero-skill team showed reduced benefits, suggesting that hetero-personality played a more central role than hetero-skill differentiation in this scenario.

The balanced personality scenario revealed strain across all configurations, with elevated stress and diminished performance and health. However, heterogeneous teams showed better outcomes than the homogeneous team, especially in cases with either hetero-personality or hetero-skill diversity as well. These findings suggest that even modest heterogeneity may provide buffering effects when teams lack pronounced stabilizing traits. The homogeneous team, by contrast, showed the steepest decline in outcomes, indicating limited adaptive capacity in uniform trait environments.

In the high extraversion and high agreeableness scenario, heterogeneous teams again displayed better than the homogeneous group, especially in the hetero-skill + personality configuration. These profiles may have promoted more effective communication and social support, which could be critical during extended confinement. Performance remained more stable, and cohesion patterns indicated stronger interpersonal alignment over time.

Team cohesion trends further differentiated team types. In the 2 × 2 heterogeneity design, teams with hetero-personality regardless of hetero-skill level showed more pronounced cohesion development across members. In contrast, teams with hetero-skill focused exhibited flatter or delayed cohesion trajectories. These differences may reflect how psychological diversity enables a broader range of relational strategies and emotional responses, facilitating more robust interpersonal adaptation under pressure.

The Monte Carlo results helped clarify not only the average differences in team outcomes, but also how much those outcomes varied across different team types. Homogeneous teams tended to be more consistent, but their results were also less favorable overall higher stress and lower performance and health. On the other hand, heterogeneous teams, particularly those with personality diversity, showed more variation in their results but generally performed better on average. This wider spread showed us again that psychological diversity may allow for more flexible and adaptive responses, even if it comes with less predictability. In high-stress, long-duration missions, having that range of potential responses could be a strength rather than a drawback.

The relationship between stress and performance remained central. Heterogeneous teams operated closer to the model's optimal stress-performance zone, while homogeneous teams were more likely to experience overstress, leading to accelerated health decline and performance degradation. These trends reinforce the importance of managing psychological load through crew composition and support structures.

These insights show the nuanced trade-offs that future crew planners may need to consider. While personality-based heterogeneity appears beneficial for resilience and performance, it also introduces variability that could impact predictability. Selection strategies might benefit from balancing diversity with targeted cohesion building approaches. Psychological complementarity, rather than mere variety, may be a more robust goal in crew assembly.

This study has several modeling limitations that should inform future extensions. Personality traits were assumed to remain fixed, omitting possible adaptation or state dependent changes over time. The cohesion mechanism was also abstracted and did not capture

leadership dynamics, interpersonal conflict, or specific mission events. Exploring more complex behavioral processes such as multi-team structures, cultural overlays, or stress-related disruptions could offer a deeper view into how teams evolve under pressure.

## 5 Conclusions

This study presented an ABM framework to examine how team composition specifically psychological and functional heterogeneity affects stress, performance, health, and cohesion during long-duration Mars mission scenarios. By simulating both homogeneous and heterogeneous teams across defined personality trait conditions, the results offer structured evidence that team diversity may contribute to greater resilience under extended isolation and operational load.

Across most scenarios, heterogeneous teams tended to outperform homogeneous teams. Teams with variation in personality traits—particularly those combining high conscientiousness with low neuroticism, or high extraversion with high agreeableness, showed lower stress levels and slightly improved performance and health outcomes. These patterns suggest that a broader mix of coping styles and interpersonal dynamics might help teams maintain stability over time. Hetero-skill diversity had a more limited effect on its own, but when paired with personality diversity, it appeared to reinforce overall outcomes.

Under balanced personality conditions, all configurations showed higher stress and steeper performance declines. However, heterogeneous teams still fared better than homogeneous ones, especially those with personality variation. This suggests that even modest trait diversity may buffer teams when no dominant stabilizing profile is present, helping preserve cohesion and delay functional degradation.

Monte Carlo analysis supported the robustness of the simulation findings. Across 1,000 randomized runs, heterogeneous teams particularly those with personality diversity tended to maintain better stress, performance, and health outcomes. These patterns held across varied starting conditions, reinforcing the idea that team diversity may help sustain mission-critical functioning even in unpredictable environments.

The results may inform future crew design strategies for long-duration space missions. Incorporating personality assessments into selection processes, and deliberately composing teams with complementary psychological and functional profiles, could improve cohesion, stress regulation, and operational stability in isolated, high-demand environments.

Future work is planned to enhance the model by introducing dynamic trait adaptation, cultural and gender dimensions, or real-time feedback loops. Expanding the simulation framework to include mission critical disruptions, resource limitations, or multi-team coordination scenarios could further improve ecological validity. Validation with data from analog environments or historical missions would help ground these insights in operational practice. The ABM developed here provides a flexible foundation for advancing our understanding of human performance in extreme, crewed spaceflight contexts.

## Author contributions

**Conceptualization:** Hao Chen.

**Data curation:** Iser Pena.

**Formal analysis:** Iser Pena.

**Funding acquisition:** Hao Chen.

**Investigation:** Iser Pena.

**Methodology:** Iser Pena.

**Project administration:** Hao Chen.

**Resources:** Hao Chen.

**Software:** Iser Pena.

**Supervision:** Hao Chen.

**Validation:** Hao Chen.

**Visualization:** Iser Pena.

**Writing – original draft:** Iser Pena.

**Writing – review & editing:** Hao Chen.

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
