## [Decision Letter · Decision Letter 0]

2 Apr 2025

PONE-D-25-00076Exploring Team Dynamics and Performance in Extended Space Missions Using Agent-Based ModelingPLOS ONE

Dear Dr. Chen,

Thank you for submitting your manuscript to PLOS ONE. After careful consideration, we feel that it has merit but does not fully meet PLOS ONE’s publication criteria as it currently stands. Therefore, we invite you to submit a revised version of the manuscript that addresses the points raised during the review process.

We look forward to receiving your revised manuscript.

Kind regards,

Peter G. Roma, Ph.D.

Academic Editor

PLOS ONE

Journal Requirements:

Reviewers' comments:

Reviewer's Responses to Questions

**Comments to the Author**

1. Is the manuscript technically sound, and do the data support the conclusions?

Reviewer #1: Partly

Reviewer #2: Partly

2. Has the statistical analysis been performed appropriately and rigorously? 

Reviewer #1: Yes

Reviewer #2: Yes

3. Have the authors made all data underlying the findings in their manuscript fully available?

Reviewer #1: Yes

Reviewer #2: Yes

4. Is the manuscript presented in an intelligible fashion and written in standard English?

Reviewer #1: Yes

Reviewer #2: Yes

5. Review Comments to the Author

Reviewer #1: Thank you for the opportunity to review Team Dynamics and Performance in Extended Space Missions Using Agent-Based Modeling. I was excited to read this manuscript and applaud the authors for attending to critical team dynamic issues related to extended space travel. The use of agent-based modeling to address the topic was commendable and, as the authors alluded to, offers to advance our understanding of psychological issues relevant to future space travel. While I applaud the general premise of the research conducted for the paper, I have some concerns about the theorizing and presentation of the work.

Literature Review

• The authors used a ‘gap argument’ to make the case for the motivation for this research (for example: “…current research lacks comprehensive models…”). Instead of pointing out the gap in the current literature, I recommend that the authors make a stronger case for the unique contribution of this research. The absence of more extensive research on team dynamics in extended space travel is likely due to a valid reason, so highlighting why this research is relevant now will help the authors highlight how they are paving the way for future studies.

• Psychological adaptation is introduced in the literature review; however, the introduction was abrupt as this construct was not previously mentioned. I see how psychological adaptation could be relevant. Still, I encourage the authors to consider how adaptation fits with the narrative about the role of the Big Five dimensions and to frame the opening of the paper with this argument to make it clear why they refer to this literature.

• What makes personality a key factor especially important in high-stress and isolated environments? The authors make this statement without explaining. I don’t disagree with the statement, but it would be helpful to explain the reasoning behind it—which would also strengthen the rationale for the paper.

• I encourage the authors to review the following to help bolster their discussion of the role of team composition in the context of space missions. While this paper is cited, there is more from this work, which primarily covers team composition— directly related to the current manuscript.

o Bell, S. T., Brown, S. G., Outland, N. B., & Abben, D. R. (2015). Critical team composition issues for long-distance and long-duration space exploration: A literature review, an operational assessment, and recommendations for practice and research.

Methodology

• In the description of the model, the heterogeneous teams varied not just by personality but also by cultural background, roles, and skills. If we see effects in the heterogeneous teams that do not replicate in the homogeneous teams, how can we be confident that the differences are due to personality? From an experimental perspective, varying cultural backgrounds, roles, and skills for the heterogeneous teams would introduce a confound. Could you explain the decision to vary other characteristics besides personality for the heterogenous teams only?

• Team cohesion is introduced as an outcome very late in the paper. It would be beneficial to introduce cohesion earlier in the paper and discuss its theoretical fit, rather than it appearing as an afterthought.

• Overall, the theoretical framework could be more focused. By honing in on the precise contribution of the study, the authors can more strongly communicate the potential impact of their research. Once the focus is clarified, the authors should revisit the literature review to align the opening with the methodology.

To reiterate, the precise focus of the study is a bit unclear. However, with some adjustments, the paper could benefit from a coherent focus and stronger theorizing about the role of crew personality composition in the context of extended space missions.

I hope the authors find my comments helpful in strengthening their paper and the overall contribution of their research.

Reviewer #2: Thank you for the opportunity to review your paper entitled “Exploring Team Dynamics and Performance in Extended Space Missions Using Agent-Based Modeling.” This paper presented an agent-based model that simulated personality composition in teams and the impact of different personality configurations on important outcomes, such as stress and performance, during a long-duration space flight. I was impressed by both the presentation and execution of this work, and I found this paper to be well-written and highly transparent (especially regarding modeling decisions). Below, I include several considerations that I hope will have a positive impact on this line of research:

1. In your literature review, you mention several studies that have simulated long-duration space flight via computational modeling. I think it would also be beneficial to include recent work by Alina Lungeanu, as she and her collaborators have discussed and published several computational models simulating teams in long-duration space flight (e.g., Lungeanu et al., 2022; Antone et al., 2023; Mesmer-Magnus et al., 2020). This may help to also bring in work from communications and the organizational sciences into your literature review so your work can speak to a wider audience.

2. I appreciate the attempt to create a figure that succinctly explains the differences between the real-world and the simulated environment of an ABM in Figure 1, but I’m not sure Figure 1 is really adding anything useful to your manuscript. Because I already understand ABMs, I see what you were trying to do with this figure, but I’m not sure someone who is unfamiliar with ABMs is going to understand what this figure means based on its current form. You might consider either revising it or dropping it from the paper, as comparatively, your other figures provide much more information than this one.

3. On page 8-9, you discuss the IMO model, which is one of the foundational models for team processes. However, in Figure 2 and in the later text on page 9, you lay out your IMO model as an “IMOI model” instead (i.e., output loops back to impact input again, and the process updates and repeats). In the teams literature, the IMOI model has largely replaced the IMO model because it captures the dynamic nature of teamwork (which is frequently not just one performance episode, but many performance episodes), and you capture this idea already in text and in your figure, but just not in your labeling and citations, so I would recommend switching to the IMOI framework for your paper (see Ilgen et al., 2005).

4. You define heterogenous teams (in Table 1) as having not only heterogeneity in terms of personality composition, but also in terms of skills/roles and task proficiency. However, in your discussion of your model results, you only mention the findings as resulting from heterogeneity in personality composition, and not the heterogeneity that was built into the model to represent different but overlapping roles and expertise among team members. However, the teams literature would suggest that heterogeneity in terms of roles and expertise is also a critical predictor of important team outcomes such as stress and performance. Are you isolating the effects of heterogeneity of personality from the effects of heterogeneity of skills/roles? If so, how? And if not, this is something that I would recommend revising in order to make sure your model results are due to your intended predictors and not extraneous influences. Perhaps you could have a 2x2 design where you test high (low) personality heterogeneity and high (low) skill/role heterogeneity to parse out the effects of each more completely.

5. Related to my last comment, how were theoverlapping skills sets (i.e., homogenous teams) versus diverse skill sets (i.e., heterogenous teams) operationalized? Perhaps I missed it, but I don’t remember seeing this explicitly outlined in your paper. Because it’s still possible, in my opinion, for these variables to have important effects in your model, this is something that would benefit from more clarity.

6. On p. 18-19, you discuss team cohesion within the model. However, some of your language suggests that individual astronauts/agents having individual “cohesion scores” (e.g., “For instance, astronaut A3 experienced a notable 684 dip in cohesion around day 167…”). Yet, cohesion is a group-level metric and not an individual-level metric, so I would clarify in this section what you mean by an individual’s cohesion. I suspect you mean to discuss their social standing with others, rather than their individual cohesion.

7. Also related to my last comment, how was cohesion operationalized? It sounds like it is a network of social ties between astronauts/agents, but are these ties directed or undirected? Is it meaningful if one astronaut says they have a positive relationship with another, and that person does not reciprocate? Adding additional details here would be helpful to make sure the reader can follow along with the details of what is happening with cohesion.

Overall, I found this paper to be quite an interesting and enjoyable read. I hope you find these comments useful as you continue to develop this work.

Works cited in this review:

Lungeanu, A., DeChurch, L. A., & Contractor, N. S. (2022). Leading teams over time through space: Computational experiments on leadership network archetypes. The Leadership Quarterly, 33(5), 101595.

Antone, B., Lungeanu, A., Bell, S. T., DeChurch, L. A., & Contractor, N. (2020). Computational modeling of long-distance space exploration: a guide to predictive and prescriptive approaches to the dynamics of team composition. In Psychology and human performance in space programs (pp. 107-130). CRC Press.

Mesmer-Magnus, J., Lungeanu, A., Harris, A., Niler, A., DeChurch, L. A., & Contractor, N. (2020). Working in space: Managing transitions between tasks. In Psychology and human performance in space programs (pp. 179-203). CRC Press.

Ilgen, D. R., Hollenbeck, J. R., Johnson, M., & Jundt, J. (2005). Teams in organizations: From I–P–O models to IMOI models. Annual Review of Psychology.

6. PLOS authors have the option to publish the peer review history of their article (what does this mean?). If published, this will include your full peer review and any attached files.

Reviewer #1: No

Reviewer #2: No

---

## [Author Response · Author response to Decision Letter 1]

8 Jun 2025

Please check the detailed response in "Response to Reviewers_PLOSONE.pdf" file attached.

---

## [Decision Letter · Decision Letter 1]

14 Aug 2025

PONE-D-25-00076R1Exploring Team Dynamics and Performance in Extended Space Missions Using Agent-Based ModelingPLOS ONE

Dear Dr. Chen,

Thank you for submitting your manuscript to PLOS ONE. After careful consideration, we feel that it has merit but does not fully meet PLOS ONE’s publication criteria as it currently stands. Therefore, we invite you to submit a revised version of the manuscript that addresses the points raised during the review process.

We look forward to receiving your revised manuscript.

Kind regards,

Peter G. Roma, Ph.D.

Academic Editor

PLOS ONE

Journal Requirements:

Additional Editor Comments (if provided):

Reviewers' comments:

Reviewer's Responses to Questions

**Comments to the Author**

1. If the authors have adequately addressed your comments raised in a previous round of review and you feel that this manuscript is now acceptable for publication, you may indicate that here to bypass the “Comments to the Author” section, enter your conflict of interest statement in the “Confidential to Editor” section, and submit your "Accept" recommendation.

Reviewer #2: (No Response)

2. Is the manuscript technically sound, and do the data support the conclusions?

Reviewer #2: Yes

3. Has the statistical analysis been performed appropriately and rigorously? 

Reviewer #2: Yes

4. Have the authors made all data underlying the findings in their manuscript fully available?

Reviewer #2: Yes

5. Is the manuscript presented in an intelligible fashion and written in standard English?

Reviewer #2: No

6. Review Comments to the Author

Reviewer #2: Thank you for the opportunity to review the revised version of your paper entitled “Exploring Team Dynamics and Performance in Extended Space Missions Using Agent-Based Modeling.” There are some significant improvements in this version of your paper. However, I have some remaining concerns that I believe need to be addressed in order to enhance the quality and deliver on the stated impact of this work. I have organized my review into two sections: addressing my previous comments and a few new comments based on your revisions to the paper.

My previous comments:

#1-3 and #6-7 have all been sufficiently addressed

Re: previous comments #4-5. Thank you for the addition of the 2x2 design examining how both heterogeneity of personality composition and heterogeneity of skills/roles differentially predict stress, performance, and health in your model. I think disentangling these effects is important for enabling mission planners to effectively select and compose their teams for long-duration space missions. I can also tell you put a lot of work into the new analyses, results, and corresponding figures. However, because this was such a big overhaul of your design and results, there needs to be more work done to guide the reader through this aspect of your paper. As of right now, the 2x2 design appears abruptly in your paper without sufficient explanation as to why it is important to examine these two types of heterogeneity. I can think of two possible ways to address this issue, but perhaps you have other ideas that are even better:

a) You could edit your abstract, intro, lit review, etc. to make the case for studying heterogeneity in two forms (personality and skills/roles), so that this extended analysis becomes a key contribution of your paper

b) You could keep the current focus on personality heterogeneity but include the analyses with the 2x2 after including a discussion about why the heterogeneity of skills/roles is an important control variable in your model and then discuss whether or not personality heterogeneity sufficiently explained variance in your outcomes above and beyond that of skills/role heterogeneity

New comments:

1. On page 2, you state that you are “explicitly modeling how psychological adaptation interacts with individual personality traits.” What exactly does this mean? The way it is phrased, I would expect to see psychological adaptation be a key variable in your model, but it is not (I think it’s more likely a potential explanatory mechanism). I would review and rephrase this to make sure it is clear what you mean.

2. Your introduction to ABM on p. 6 comes very abruptly with no transition from the previous section. In your writing, you first need to make the case for ABM and its benefits for investigating these research questions before you introduce the technicalities of ABM in section 1.1.3.

3. You mention on P. 8 that ABMs can be used by mission planners to model diverse agents etc., but I highly doubt mission planners are loading up an ABM to test theses team composition questions. Instead, they likely review research (such as yours) that used an ABM along with their own knowledge to make decisions. Perhaps consider rephrasing (unless you know that they are definitely using ABMs like this to make their team composition decisions).

4. Should Figure 1 say personal traits or personality traits? Why is cultural adaptation underlined?

5. The role specialization portion of your paper (beginning in section 2.3) comes out of nowhere for a normal reader since they are not privy to these reviews. This is also not mentioned in your intro or abstract, or really anywhere else. However, the reason you have to separate out these effects is because your “story” was about how heterogeneity of teams impacts team success.

6. Your discussion of how skill heterogeneity is included in your model (bottom of page 12 and top of page 13) does not match your current 2x2 configuration. Please be sure to re-read your entire manuscript in light of the changes you have made to ensure that your descriptions still make sense, because right now these descriptions are not in alignment with this version of your model.

7. In Figure 7, are you still showing “stress, performance, and health”? There are no labels for these in the figure.

This is a much improved version of your paper, and I hope these additional comments enable you to boost the quality of this work even more.

7. PLOS authors have the option to publish the peer review history of their article (what does this mean?). If published, this will include your full peer review and any attached files.

Reviewer #2: No

---

## [Author Response · Author response to Decision Letter 2]

29 Aug 2025

Please see attached "Response to Reviewers_PLOSONE_R2" file.

---

## [Editor Report · Decision Letter 2]

2 Sep 2025

Exploring Team Dynamics and Performance in Extended Space Missions Using Agent-Based Modeling

PONE-D-25-00076R2

Dear Dr. Chen,

We’re pleased to inform you that your manuscript has been judged scientifically suitable for publication and will be formally accepted for publication once it meets all outstanding technical requirements.

Kind regards,

Peter G. Roma, Ph.D.

Academic Editor

PLOS ONE
---

## [Editor Report · Acceptance letter]

PONE-D-25-00076R2

PLOS ONE

Dear Dr. Chen,

I'm pleased to inform you that your manuscript has been deemed suitable for publication in PLOS ONE. Congratulations! Your manuscript is now being handed over to our production team.

Kind regards,

on behalf of

Dr. Peter G. Roma

Academic Editor

PLOS ONE